# AMPK modulation ameliorates dominant disease phenotypes of *CTRP5* variant in retinal degeneration

Kiyoharu J. Miyagishima [1,10], Ruchi Sharma[2,10], Malika Nimmagadda[2,10], Katharina Clore-Gronenborn[2], Zoya Qureshy[2], Davide Ortolan [2], Devika Bose[2], Mitra Farnoodian[2], Congxiao Zhang[1], Andrew Fausey[2], Yuri V. Sergeev[3], Mones Abu-Asab[4], Bokkyoo Jun[5], Khanh V. Do[5], Marie-Audrey Kautzman Guerin[5], Jorgelina Calandria[5], Aman George[3], Bin Guan[6], Qin Wan[1], Rachel C. Sharp [7], Catherine Cukras[8], Paul A. Sieving [9], Robert B. Hufnagel[6], Nicolas G. Bazan [5], Kathleen Boesze-Battaglia[7], Sheldon Miller[1] & Kapil Bharti [2✉]

Late-onset retinal degeneration (L-ORD) is an autosomal dominant disorder caused by a missense substitution in *CTRP5*. Distinctive clinical features include sub-retinal pigment epithelium (RPE) deposits, choroidal neovascularization, and RPE atrophy. In induced pluripotent stem cells-derived RPE from L-ORD patients (L-ORD-iRPE), we show that the dominant pathogenic *CTRP5* variant leads to reduced CTRP5 secretion. *In silico* modeling suggests lower binding of mutant CTRP5 to adiponectin receptor 1 (ADIPOR1). Downstream of ADIPOR1 sustained activation of AMPK renders it insensitive to changes in AMP/ATP ratio resulting in defective lipid metabolism, reduced Neuroprotectin D1(NPD1) secretion, lower mitochondrial respiration, and reduced ATP production. These metabolic defects result in accumulation of sub-RPE deposits and leave L-ORD-iRPE susceptible to dedifferentiation. Gene augmentation of L-ORD-iRPE with WT CTRP5 or modulation of AMPK, by metformin, re-sensitize L-ORD-iRPE to changes in cellular energy status alleviating the disease cellular phenotypes. Our data suggests a mechanism for the dominant behavior of *CTRP5* mutation and provides potential treatment strategies for L-ORD patients.

[1] Section on Epithelial and Retinal Physiology and Disease, NEI, NIH, Bethesda, MD 20892, USA. [2] Ocular and Stem Cell Translational Research Section, NEI, NIH, Bethesda, MD 20892, USA. [3] Ophthalmic Genetics and Visual Function Branch, National Eye Institute, NIH, Bethesda, MD 20892, USA. [4] Section of Histopathology, National Eye Institute, NIH, Bethesda, MD 20892, USA. [5] Neuroscience Center of Excellence, Louisiana State University Health, New Orleans, LA 70112, USA. [6] Medical Genetics and Ophthalmic Genomics Unit, NEI, NIH, Bethesda, MD 20892, USA. [7] Department of Biochemistry University of Pennsylvania, 240 South 40th Street, Levy Building, Room 515, Philadelphia, PA 19104, USA. [8] Division of Epidemiology and Clinical Applications and Ophthalmic Genetics and Visual Function Branch, NEI, NIH, Bethesda, MD 20892, USA. [9] Section for Translation Research in Retinal and Macular Degeneration, NEI, NIH, Bethesda, MD 20892, USA. [10]These authors contributed equally: Kiyoharu J. Miyagishima, Ruchi Sharma, Malika Nimmagadda. ✉email: kapil.bharti@nih.gov

Late-onset retinal degeneration (L-ORD) is a rare, genetically dominant monogenic retinal dystrophy that is characterized by atrophic or neovascular pathology of the retinal pigment epithelium (RPE)[1]. RPE is a monolayer of polarized, pigmented cells at the back of the eye that plays a critical role in maintaining the homeostasis of the subretinal space and adjacent photoreceptor cells necessary for vision[2]. L-ORD-induced retinal changes are minimal until after 40 years of age[3]. Initial evidence of disease includes the presence of reticular pseudodrusen and thinning of the choroid, and later in life, progresses to advanced stages characterized by increased sub-RPE deposits, extrafoveal RPE atrophy, and choroidal neovascularization[4,5]. Thus, we hypothesized that an analysis of L-ORD-retinal pigment epithelium derived from a family with the p.Ser163Arg mutation could provide insight into the cellular mechanisms underlying its pathogenesis[6].

L-ORD is caused by amino acid substitutions (e.g., p.Ser163Arg) in the CTRP5 protein, encoded by the C1QTNF5 gene, a bicistronic mRNA partner of membrane frizzled-related protein (MFRP)[7]. The C1q/TNF-related protein (CTRP) family are paralogues to ADIPONECTINS[8], and both protein families are widely studied for their roles in regulating energy homeostasis and fatty acid metabolism in non-eye tissues[8,9]. Like ADIPONECTIN, all CTRP members possess four distinct structural domains: (1) N-terminal signal peptide that targets the protein for secretion; (2) variable region; (3) collagen region; and (4) a C-terminal globular domain[10]. All CTRPs form bouquet-like trimeric structures, and some family members, including CTRP5, can assemble into homologous or heterologous higher-order multimeric complexes[10–12]. Adiponectin proteins affect cellular metabolism through adiponectin receptors 1 (ADIPOR1) and ADIPOR2[13,14]. Recent publications suggest that a member of the CTRP family of proteins, CTRP9, also acts as a ligand for ADIPOR1 and ADIPOR2[15,16]. ADIPOR1 primarily regulates gluconeogenesis and fatty acid oxidation, whereas ADIPOR2 is mainly involved in oxidative stress and inflammation[17].

Previously, mutant CTRP5 was shown to form heterooligomers with wild-type CTRP5[11,18,19], but the mechanism for genetically dominant behavior of CTRP5 mutations is still not clear. CTRP5 protein has been identified as a putative biomarker for obesity and chronic obstructive pulmonary disease[20], suggesting a role for this protein in regulating cellular fatty acid metabolism. The 5′ AMP-activated protein kinase (AMPK) has been suggested as an intracellular mediator of CTRP5 to regulate fatty acid metabolism and energy homeostasis[21]. RPE cells participate in diurnal phagocytosis of photoreceptor outer segments (POS) which have an abundance of fatty acids and lipids[22,23]. Increasing evidence suggests that RPE cells metabolize fatty acids from POS membranes and recycle metabolic substrates to photoreceptors through ketogenesis. RPE also secrete docosahexaenoic acid (DHA)-derived neuroprotective factors like Neuroprotectin D1 (NPD1) that protects the photoreceptors from photooxidative stress and, in a cell-autonomous fashion, the RPE itself[23–26]. AMPK has been suggested as a key regulator of POS digestion in the RPE[27]. It has been suggested that dysregulation of fatty acid and lipid metabolism contributes to the formation of sub-RPE deposits in AMD[28].

In the present study, we demonstrated that L-ORD RPE derived from patient induced pluripotent stem cells (L-ORD-iRPE) accurately recapitulate the human disease phenotype: elevated sub-RPE deposition of APOE - a demonstrated component of drusen, and mispolarized secretion of vascular endothelial growth factor (VEGF) - a causative factor of CNV. Mechanistically, reduced secretion of CTRP5 and predicted lower binding affinity of mutant CTRP5 to ADIPOR1 receptor is the likely reason for the genetically dominant behavior of this disease. We show that lower CTRP5 levels are associated with constitutively activated AMPK leading to its insensitivity to changes in the cellular energy status. Using a gene therapy approach, overexpression of WT CTRP5 in patient cells overcomes lower CTRP5 levels and rescue mispolarized VEGF secretion. Metformin, an anti-diabetic drug, rescued L-ORD-iRPE metabolic dysfunction by resensitizing AMPK to changes in cellular stress, restoring energy homeostasis, and ameliorating disease cellular phenotypes.

## Results

Induced pluripotent stem cells (iPSCs) were derived from fibroblasts isolated from skin biopsies of four siblings—two had a clinical and molecular diagnosis of L-ORD due to the recurrent p.Ser163Arg variant, and the other two were unaffected and did not carry the pathogenic variant[7]. All iPSC lines expressed pluripotency markers: OCT4, NANOG, SOX2, and SSEA4 and were karyotypically normal (Table 1 and Fig. S1). An in vitro embryoid body assay demonstrated similar capabilities between all iPSC clones to differentiate into cell types from all three germ layers (Table 1). iRPE derived from two iPSC clones per donor were used for further experiments. Thus, each experiment presented here uses averaged data from iRPE derived from four iPSC clones of two unaffected siblings (healthy-iRPE) and four iPSC clones of two patients (L-ORD-iRPE).

**L-ORD pathophysiology replicated in patient-derived iRPE cells.** C1QTNF5 (NM_001278431.2) exon 2 was Sanger sequenced in iPSCs to verify that only the patient cells retained the S163R L-ORD mutation (Fig. 1a). All eight iPSC clones were differentiated into RPE cells and matured on transwells for 6 weeks to

**Table 1 Validation of pluripotency, presence of pathogenic C1QTNF5 variant, karyotyping was performed on iPSCs in the study. Identity (short tandem repeat) analysis was performed on iRPE and iPSCs to ensure no cross contamination.**

| Donor derived cells | iPSC line clone | Pluripotency markers | Germ-layer markers | Normal karyotype | Sanger sequencing | Short tandem repeat |
|---|---|---|---|---|---|---|
| Healthy-iRPE Donor 1 | LORCF1-Clone1 | ✓ | ✓ | ✓ | ✓ | ✓ |
| Healthy-iRPE Donor 1 | LORCF1-Clone2 | ✓ | ✓ | ✓ | ✓ | ✓ |
| Healthy-iRPE Donor 2 | LORCF2-Clone1 | ✓ | ✓ | ✓ | ✓ | ✓ |
| Healthy-iRPE Donor 2 | LORCF2-Clone2 | ✓ | ✓ | ✓ | ✓ | ✓ |
| L-ORD-iRPE Donor 1 | LORPF1-Clone1 | ✓ | ✓ | ✓ | ✓ | ✓ |
| L-ORD-iRPE Donor 1 | LORPF1-Clone2 | ✓ | ✓ | ✓ | ✓ | ✓ |
| L-ORD-iRPE Donor 2 | LORPF2-Clone1 | ✓ | ✓ | ✓ | ✓ | ✓ |
| L-ORD-iRPE Donor 2 | LORPF2-Clone2 | ✓ | ✓ | ✓ | ✓ | ✓ |

Miyagishima et al. investigate the pathological mechanisms underlying mutant CTRP5 function in late-onset retinal degeneration (L-ORD). With human iPSC-derived RPE cells, they demonstrate that in L-ORD-iRPE, constitutive activation of AMPK disrupts cellular metabolism/energy homeostasis, changes apical/basal VEGF secretion, and metformin treatment corrects these associated phenotypes.

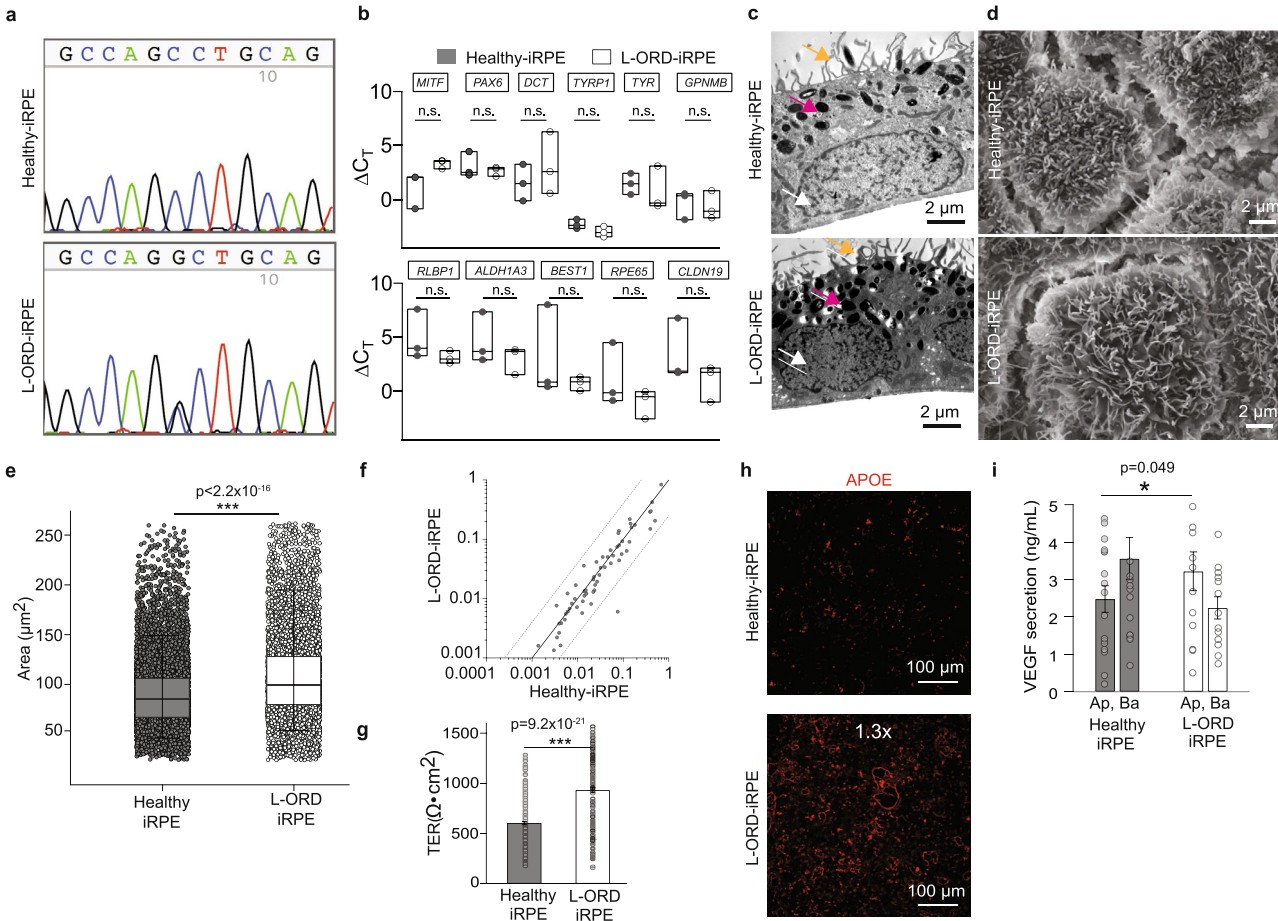

**Fig. 1 L-ORD-iRPE replicates disease cellular phenotype in a dish. a** Sanger sequencing confirms the presence of p.Ser163Arg variant in L-ORD-iPSCs. The heterozygous single nucleotide variant in *C1QTNF5* (c.489 C > C/G) appears as a peak within a peak. **b** Boxplots of ΔCt values for RPE signature genes are comparable in healthy and L-ORD-iRPE (boxplots represent $n = 3$ independent experiments from at least two donors; 25th, median, and 75th percentiles correspond to the bottom, middle, and top of each box). **c** TEM images of healthy and L-ORD-iRPE monolayers show polarized RPE structure including abundant apical processes (orange arrow), melanosomes (magenta arrow), and basally located nuclei (white arrow). Scale bar: 2 μm. **d** SEM images of healthy-iRPE and L-ORD-iRPE demonstrate hexagonal morphology and abundant apical processes. Scale bar: 2 μm. **e** Boxplot of cell area reveals larger cell size in L-ORD-iRPE than heathy-iRPE. iRPE monolayers were immunostained with membrane marker (ADIPOR1) to outline their hexagonal shape (healthy-iRPE: $n = 9$; L-ORD-iRPE: $n = 6$). Total of $n > 10,000$ cells counted. **f** Scatterplot of dedifferentiation-related genes shows no difference in L-ORD-iRPE ($n = 2$ donors) compared to healthy-iRPE ($n = 2$ donors). **g** Transepithelial resistance (TER) measurements were performed using an epithelial voltohmmeter in L-ORD-iRPE ($n = 264$ inclusive of all clones, two donors) compared to healthy-iRPE ($n = 266$ inclusive of all clones, 2 donors). (**h**) Comparative analysis of sub-RPE APOE (red)-positive deposits ($n = 25$) in healthy-iRPE ($n = 11$) and L-ORD-iRPE ($n = 14$) show 1.3-fold ($p = 0.01$) increase in APOE staining. Hoechst 33342 staining confirms the lack of cellular debris contributing towards APOE staining. Scale bar: 100 μm. **i** Apical and basal VEGF secretion is mispolarized in L-ORD-iRPE (white, $n = 13$) compared with healthy-iRPE (gray, $n = 17$). For validation of iPSC-derived lines and karyotyping analysis: See Table 1 and Fig. S1. *$p < 0.05$; ***$p < 0.001$; ns not significant.

form physiologically relevant epithelial monolayers using our previously published protocol[29]. L-ORD and healthy-iRPE monolayers expressed similar levels of developmental (*MITF*, *PAX6*, *DCT*, *TYRP1*, *TYR*, *GPNMB*) and mature RPE markers (*RLBP1*, *ALDH1A3*, *BEST1*, *RPE65*, *CLDN19*, *EZRIN*, *MFRP*)[29] (Fig. 1b and Supplementary Fig. 2a), suggesting the ability of L-ORD-associated cells to mature and acquire RPE cell-specific transcriptional profile. Transmission electron microscopy (TEM) of iRPE monolayers revealed typical polarized RPE features like abundant apical processes (orange arrow), apically located melanosomes (magenta arrow), and basally located nuclei (white arrow) in all eight iRPE samples (Fig. 1c). Scanning electron microscopy (SEM) also confirmed abundant apical processes in both healthy and L-ORD-iRPE but suggested increased cell size of L-ORD-iRPE cells as compared to healthy-iRPE (Fig. 1d and Supplementary Fig. 2b). To further confirm this observation, iRPE monolayers stained for cell borders were analyzed to assess

differences in cell size. Cell area was 35% larger and more variable in L-ORD-iRPE compared to healthy-iRPE (average cell area ±SD: $106.9 \pm 68.0$ μm² for L-ORD-iRPE and $79.2 \pm 37.2$ μm² for healthy-iRPE; $p < 0.001$) (Fig. 1e). Similar spatial irregularities have been reported in RPE cells of AMD donor eyes and are often associated with RPE cell dedifferentiation[30]. However, no significant difference in the expression of 84 dedifferentiation genes was seen in L-ORD-iRPE as compared to healthy-iRPE (Fig. 1f and Supplementary Table 1). Transepithelial resistance (TER) measurements confirmed the formation of functional tight junctions between neighboring RPE cells and showed a significant increase in L-ORD-iRPE compared to healthy-iRPE (mean ± SEM, TER: $931.3 \pm 15.6$ Ω•cm² for L-ORD-iRPE, and $602.6 \pm 15.6$ Ω•cm² for healthy-iRPE; Fig. 1g, $p < 0.001$). Sub-RPE deposits are a characteristic feature of L-ORD, which shares some similarities to the lipid-containing deposits seen in AMD eyes[31]. Apolipoprotein E (APOE), associated with high-density

lipoproteins (HDLs), is a well-known marker for sub-RPE deposits[32]. About 1.3-fold higher subcellular APOE-positive deposits were observed in L-ORD-iRPE as compared to healthy-iRPE (Fig. 1h and Supplementary Fig. 2c, $p < 0.01$). Despite normal epithelial monolayer formation, L-ORD-iRPE exhibited mispolarized vascular endothelial growth factor (VEGF) secretion (Fig. 1i). In particular, the basal VEGF secretion was reduced by 63% ($p < 0.05$) in L-ORD-iRPE compared to healthy-iRPE perhaps contributing to the thinning of the choroid observed clinically in L-ORD patients. These results suggest that the iRPE recapitulated L-ORD phenotypes associated with the most common disease-causing variant in *CTRP5*.

**Reduced CTRP5 secretion and lower binding of mutant CTRP5 with adiponectin receptor 1**. Since *C1QTNF5* and *MFRP* share a bicistronic transcript, we investigated whether the L-ORD p.Ser163Arg variant altered the expression of either transcript. qRT-PCR revealed comparable ΔCt values for *C1QTNF5* and

*MFRP* transcripts across all samples, indicating no patient-specific differences in mRNA expression ($p = 0.8$ and 0.5 for *CTRP5* and *MFRP*, respectively, Supplementary Fig. 3a). ELISA-based quantitative analysis showed L-ORD-iRPE CTRP5 apical and basal secretion was significantly lower compared to the healthy-iRPE (Ap: 14.3-fold, $p < 0.001$; Ba: 19.7-fold, $p < 0.001$, L-ORD-iRPE vs healthy) in agreement with previously reported findings[19,33] (Fig. 2a, b). These results suggested that mutant CTRP5 protein might be trapped inside L-ORD-iRPE. To determine if mutant CTRP5 traps the WT version inside the cell we used lentivirus technology to overexpress V5-tagged mutant and FLAG-tagged WT CTRP5 proteins in healthy-iRPE cells (Fig. 2c, Supplementary Fig. 3b). At a multiplicity of infection (MOI) of 0.5, expression of mutant and WT CTRP5 was barely detectable inside the cell. When the MOI of the mutant protein was increased to 3.0 while keeping the WT MOI constant at 0.5, significantly higher levels of WT CTRP5 could be detected inside the cell (Fig. 2c, compare green signal between the top and

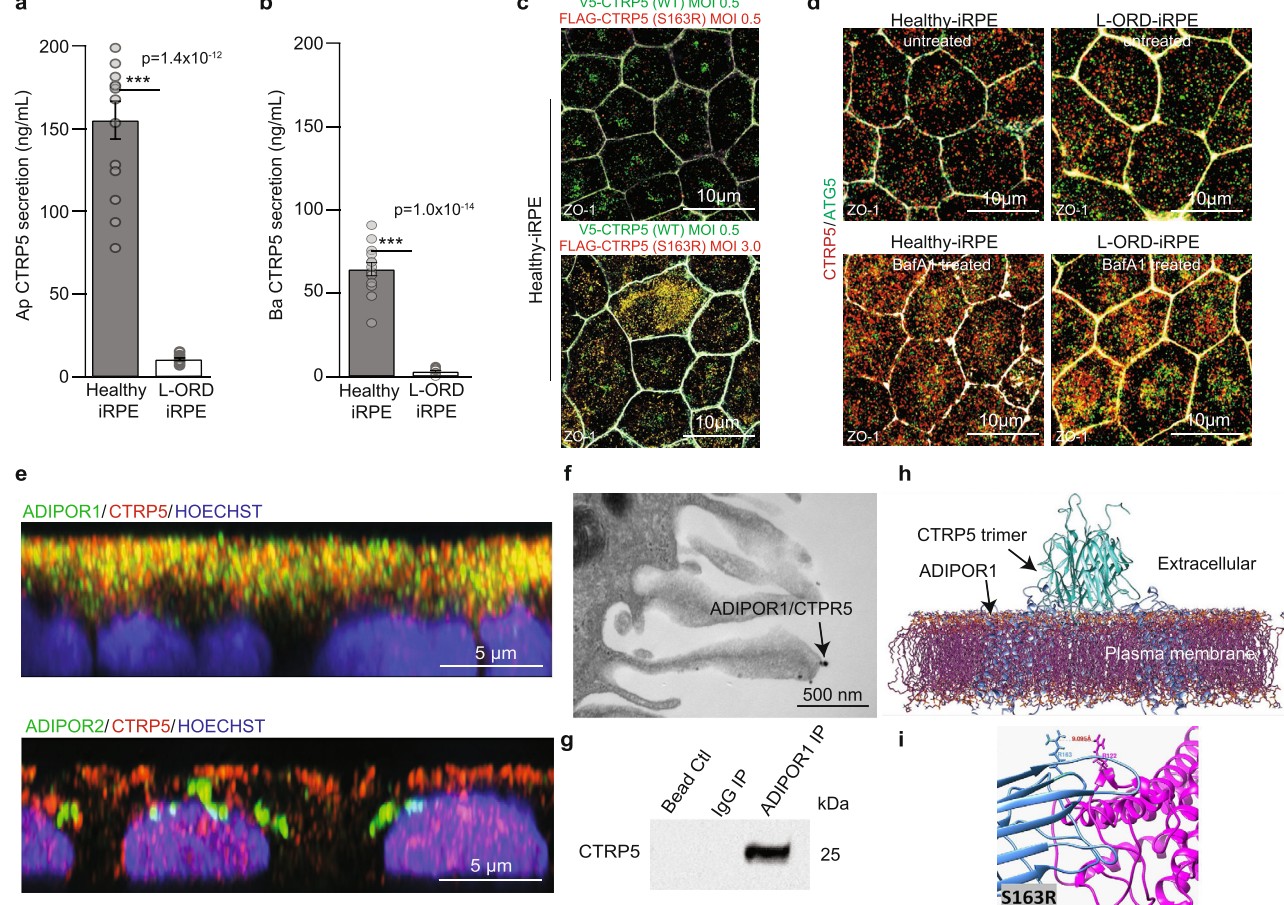

**Fig. 2 Expression and localization of CTRP5 in L-ORD-iRPE. a, b** Apical and basal CTRP5 secretion measured by ELISA in the culture medium are both significantly decreased ($n = 14$). **c** Coexpression of V5-tagged WT CTRP5 (green) and FLAG-tagged S163R CTRP5 (red) in healthy-iRPE. V5-tagged WT CTRP5 expressing lentivirus construct was transduced at MOI 0.5 for both top and bottom panels. MOI of Flag-tagged S163R CTRP5 expressing lentivirus construct was 0.5 for cells in the top panel and 3.0 for cells in the bottom panel. Scale bar: 10 μm **d** Confocal microscopy images of untreated and bafilomycin (BafA1) treated (3 h) healthy-iRPE and L-ORD-iRPE co-stained with CTRP5 (red) and ATG5 (green). ($n = 3$ images per condition). Scale bar: 10 μm. **e** Representative confocal microscopy images showing colocalization of CTRP5 (red) with membrane receptors ADIPOR1 (green, upper panel) and no colocalization with ADIPOR2 (green, lower panel). Nuclear stain (blue). Scale bar: 5 μm. **f** TEM image of immunogold labeled ADIPOR1 (6 nm gold particle) and CTRP5 (12 nm gold particle) demonstrate the co-binding of two proteins (arrow). Scale bar: 500 nm. **g** Western blot detects CTRP5 in the membrane fraction of iRPE cells immunoprecipitated using anti-ADIPOR1 antibodies and not in lanes immunoprecipitated with the IgG antibody or with only beads. **h** Probabilistic model of the interaction of integral membrane protein ADIPOR1 (blue) and CTRP5 (teal) determined using published crystallographic structures and refined by molecular dynamics. **i** The polar serine to arginine substitution on CTRP5 (blue) is predicted to have an electrostatic repulsive interaction with neighboring arginine residue, Arg122, on ADIPOR1 (magenta) reducing the likelihood of interaction. See also Fig. S4. ***$p < 0.001$.

bottom panels). Whereas overexpression of WT CTRP5 alone at an MOI of 3.0 did not cause similar accumulation inside the cell (Supplementary Fig. 3b). This suggests that the mutant protein traps the WT version inside the cell in heterooligomers and reduces its secretion (Fig. 2c). These results provide evidence that the CTRP5 pathogenic variant affects the secretion of the native wild-type protein as well.

To determine where the mutant CTRP5 is trapped inside the cell, we checked its co-localization with different intracellular organelles. Previously it was suggested that mutant CTRP5 overexpressed in HEK293 cells colocalizes with the endoplasmic reticulum[19]. In both healthy and L-ORD-iRPE, we noted similar colocalization of CTRP5 with the endoplasmic reticulum (ER) marker, CALRETICULIN, suggesting that the mutant CTRP5 is not trapped in the ER [Supplementary Fig. 3c; Pearson's correlation coefficient between CTRP5 (red) and CALRETICU-LIN (green): healthy-iRPE $0.37 \pm 0.06$; L-ORD-iRPE $0.26 \pm 0.05$. $p =$ ns]. Consistently, mRNA expression of ER-stress markers did not appear to be elevated in L-ORD-iRPE, suggesting no ER stress in L-ORD-iRPE (Supplementary Fig. 3d). Similarly, no significant difference in colocalization was noted between healthy and L-ORD-iRPE for early endosomal marker EEA1[34] [Supplementary Fig. 3e; Pearson's correlation coefficient between CTRP5 (green) and EEEA1 (red): healthy-iRPE $0.21 \pm 0.03$; L-ORD-iRPE $0.22 \pm 0.02$. $p =$ ns]. Rather, it appears that in L-ORD-iRPE, CTRP5 was targeted for lysosomal degradation (Supplementary Fig. 3f and Fig. 2d). Bafilomycin A1 (BafA1) treatment, which prevented lysosomal acidification[35] enhanced CTRP5 (red) accumulation, more so in L-ORD-iRPE as compared to healthy-iRPE (compare untreated samples with BafA1 treated samples — Supplementary Fig. S3f and Fig. 2d). Furthermore, strong colocalization was seen between CTRP5 and LAMP1 (green, lysosomal marker; Supplementary Fig. 3f) and CTRP5 and ATG5 (green, autophagosome marker; Fig. 2d). Pearson's correlation coefficient showed between 2–2.5-fold higher colocalization between CTRP5 and ATG5 in L-ORD-iRPE as compared to healthy-iRPE (Fig. 2d; untreated: healthy-iRPE $0.07 \pm 0.02$; L-ORD-iRPE $0.16 \pm 0.04$. $p < 0.05$; bafilomycin A1 treated: healthy-iRPE $0.16 \pm 0.02$; L-ORD-iRPE $0.37 \pm 0.08$. $p < 0.01$). No localization for mutant was seen in the sub-RPE space (Supplementary Fig. 3g). Overall, these results suggest that in L-ORD-iRPE, the mutant CTRP5 variant along with WT CTRP5 are degraded via the lysosomal-autophagy pathway resulting in reduced secretion.

To discover intracellular signaling pathways affected by mutant CTRP5, we looked for its potential partner receptors on the RPE cell surface. Since the globular domain of CTRP5 is ~40% homologous to ADIPONECTIN[36], we sought to determine if it interacts with the adiponectin receptor family[13]. Consistent with the dominant apical secretion of CTRP5, ADIPOR1, and ADIPOR2 receptors are predominantly present on the apical surface of healthy and L-ORD-iRPE cells (Supplementary Fig. 4a, b). Interestingly, ADIPOR1 mRNA (Supplementary Fig. 4c, $p < 0.05$) and protein (Supplementary Fig. 4d, e, $p < 0.05$) expression were found to be approximately twofold higher in healthy-iRPE compared to L-ORD-iRPE. High-resolution immunofluorescence microscopy in healthy-iRPE revealed specific co-labeling of CTRP5 (red-ligand) with Adiponectin receptor 1 (ADIPOR1, green-receptor), and not with Adiponectin receptor 2 (ADIPOR2, green) (Fig. 2e). Native immunogold labeling of healthy-iRPE further confirmed CTRP5 (12 nm gold particle) and ADIPOR1 (6 nm gold particle) interaction, as indicated by black arrows (Fig. 2f). These co-labeling experiments suggest that apically secreted CTRP5 interacts with ADIPOR1 on the RPE surface and may modulate its activity. Co-IP revealed a direct interaction between ADIPOR1 and CTRP5 (Fig. 2g) further validating microscopy findings.

To further support that native CTPR5 interacts with ADIPOR1, we modeled in silico interactions using existing structural models for both proteins[11,18,37]. Figure 2h shows a simplified representation of the heterocomplex of two structures refined in water, demonstrating that a CTRP5 trimer can be docked to the binding domain of ADIPOR1 positioned in a lipid membrane (Fig. 2h and Supplementary Methods) and may compete with ADIPOR1 ligand ADIPONECTIN for the receptor's binding cavity. Tu and Palczewski previously showed that the pathogenic p.Ser163Arg substitution alters the charge at the top of the trimeric head of CTRP5[18]. Consistent with those observations, simulation of CTRP5 binding to ADIPOR1 reveals that the Arg163 residue lies on the interface of CTRP5 expected to dock with ADIPOR1 and predicts amino acid residues that are most likely to show interaction (Fig. 2i). The serine to arginine change in CTRP5 adds a positive charge that repels the Arg122 on ADIPOR1's surface, increasing their $C\alpha$-$C\alpha$ distance by 87.5% (native: ~4.8 Å; mutant: 9.0 Å). Similar changes are predicted for other recently reported disease-associated CTRP5 variants (p.Gly216Cys, p.Prp188Thr) that are also close to its surface and are expected to repel ADIPOR1, reducing the likelihood of interaction compared to the WT protein (Supplementary Fig. 4f, g). Overall, these results suggest that the mutant protein not only reduces the secretion of the WT counterpart but it likely also reduces its binding affinity to the ADIPOR1 receptor.

**CTRP5 fine-tunes AMPK sensitivity to cellular energy status.** Adiponectin and its receptors regulate lipid metabolism in an AMP-activated protein kinase (AMPK) dependent mechanism, and the activation of ADIPOR1 stimulates ceramidase activity promoting cell survival[38]. Reduced ADIPOR1 activity leads to the accumulation of ceramides and their metabolic derivatives, which have been linked to macular degeneration[39]. Compared to healthy-iRPE, L-ORD-iRPE did not show excessive ceramide accumulation or altered ceramidase activity (Supplementary Fig. 5a, b), suggesting that lower binding of mutant CTRP5 to ADIPOR1 leads to its constitutive activation rather than inhibition of ADIPOR1 ceramidase activity. Therefore, the following experiments sought to determine whether ADIPOR1 downstream signaling, including the AMPK pathway, was constitutively active in L-ORD-iRPE.

Consistent with constitutively active ADIPOR1 in L-ORD-iRPE, at baseline, in serum-containing media, the levels of phospho (T172)-AMPK (pAMPK), a measure of AMPK activity, was ~20% higher in L-ORD-iRPE compared to healthy-iRPE ($121 \pm 7.5\%$ vs. $100 \pm 4\%$; Fig. 3a, $p < 0.01$). Since CTRP5 and ADIPOR1's natural ligand, ADIPONECTIN, are known to circulate at high concentrations in the serum[36], we hypothesized that serum-free media would uncover the effect of exogenous CTRP5 on ADIPOR1. Thus, iRPE were incubated with recombinant CTRP5 globular form (0.2 µg/mL gCTRP5) for 30 min in the presence (+) and absence (−) of serum to evaluate its role in AMPK signaling (Fig. 3b). Based on this notion, the addition of gCTRP5 to healthy or L-ORD-iRPE in serum-containing media did not alter AMPK activity ($1.1 \pm 0.09$ – L-ORD-iRPE; $1.0 \pm 0.01$ – healthy-iRPE, Fig. 3b, $p =$ ns), but in serum-deprived media, the addition of gCTRP5 led to a 20% decrease in pAMPK levels in healthy-iRPE. In contrast, this decrease was not seen in L-ORD-iRPE ($1.0 \pm 0.1$ – L-ORD-iRPE, $p =$ ns; $0.8 \pm 0.04$ – healthy-iRPE, $p < 0.01$; Fig. 3b). To better elucidate the ADIPOR1-CTRP5 (receptor-ligand) interaction, we performed a ligand dose-response curve (in serum-free media) by adding increasing concentrations of recombinant full-length CTRP5 (Fig. 3c). In healthy-iRPE, increasing the concentration of exogenous CTRP5 resulted in a dose-dependent reduction in

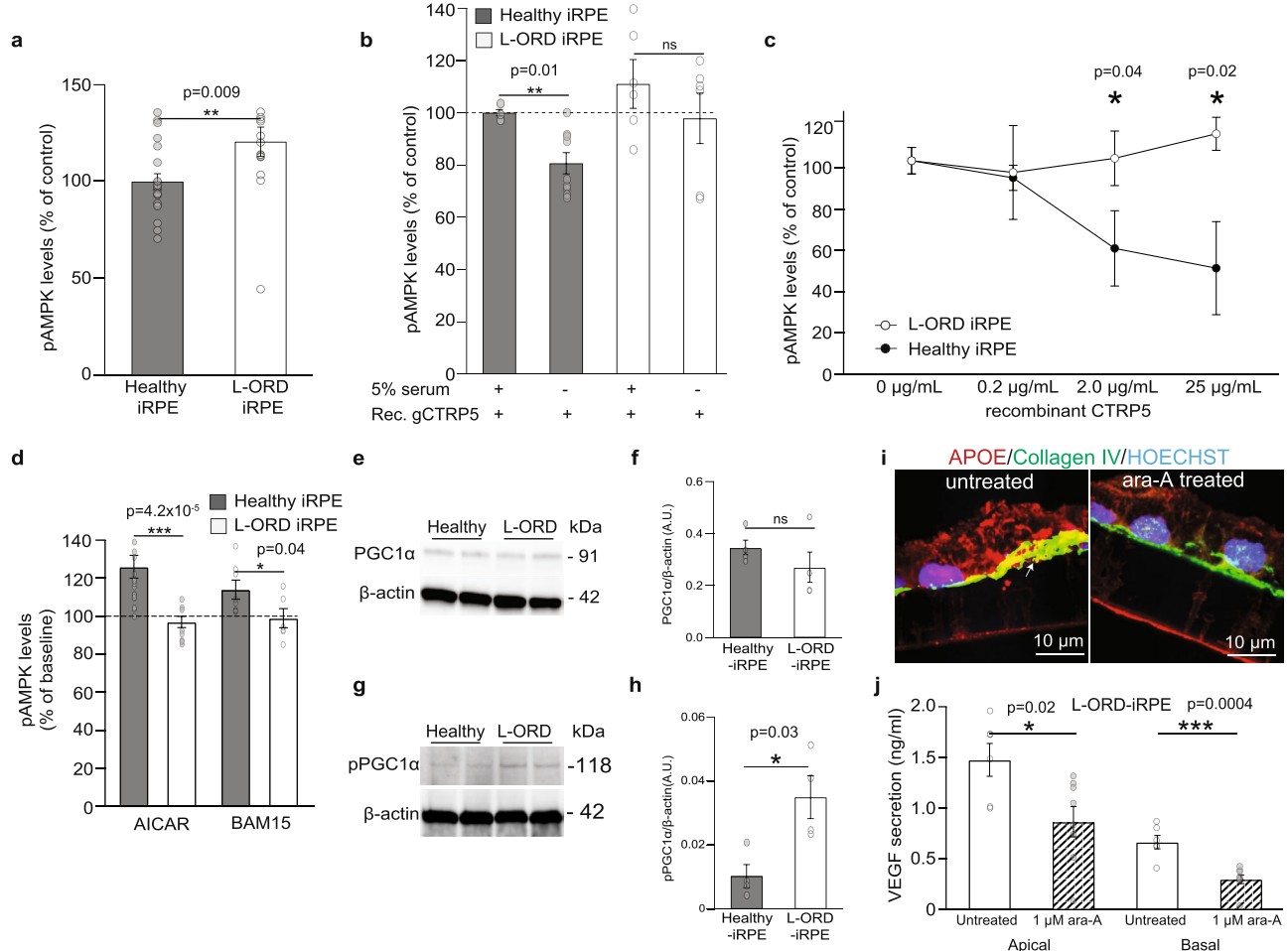

**Fig. 3 Reduced antagonism of CTRP5 on ADIPOR1 results in altered AMPK signaling in L-ORD-iRPE. a** pAMPK levels determined by ELISA in L-ORD-iRPE ($n = 15$; 120.6% ± 0.08) compared to healthy-iRPE ($n = 21$; 100% ± 0.04), cultured in 5% serum-containing media. **b** Effect of recombinant globular CTRP5 (0.2 µg/mL) on pAMPK levels in the presence (+) or absence (−) of serum in healthy-iRPE (81% ± 4, $n = 9$) and L-ORD-iRPE (99% ± 1, $n = 6$), measured by ELISA. Data were normalized to the untreated condition (0 µg/mL gCTRP5), considered as 100%. **c** Effects of recombinant full-length CTRP5 (0.2, 2, and 25 µg/mL) on pAMPK levels in healthy-iRPE ($n = 6$) and L-ORD-iRPE ($n = 6$) incubated in serum-free medium for 5 h, measured by ELISA. **d** Effect of increasing cytosolic AMP with AICAR (an AMP analog, 2 mM) treatment or decreasing ATP with BAM15 (a mitochondrial uncoupler, 500 nM) on AMPK activity in healthy-iRPE and L-ORD-iRPE. All data were normalized to the 0% serum-containing untreated condition (AICAR: $n = 20$ for healthy-iRPE, $n = 16$ for L-ORD-iRPE; BAM15: $n = 8$ for healthy-iRPE, and $n = 6$ for L-ORD-iRPE). **e** Representative Western blot of PGC1α expression in healthy and L-ORD-iRPE. **f** Quantification of Western blots for PGC1α. Healthy-iRPE ($n = 4$). L-ORD-iRPE ($n = 4$). **g** Representative Western blot for phospho (S570)-PGC1α in L-ORD-iRPE compared to healthy-iRPE. **h** Quantification of Western blots for phosphor (S570)-PGC1 α in healthy-iRPE ($n = 4$) and L-ORD-iRPE ($n = 4$). **i** APOE expression (red) in untreated and adenine 9-beta-d-arabinofuranoside (1 µM, ara-A) treated L-ORD-iRPE. Collagen IV (green) marks RPE basal surface and nuclei (blue) are labeled with Hoechst 33342. ($n = 3$ images each) Scale bar: 10 µm. **j** ELISA detection of apical and basal VEGF secretion in untreated and adenine 9-beta-d-arabinofuranoside (1 µM, ara-A) treated L-ORD-iRPE ($n = 6$). ∗$p < 0.05$; ∗∗$p < 0.01$; ∗∗∗$p < 0.001$; ns not significant, AU arbitrary units.

pAMPK levels at (50% reduction at 25 µg/mL, $p < 0.05$). But in L-ORD-iRPE, the addition of CTRP5 did not affect AMPK activity, suggesting that the mutant CTRP5 has a dominant effect on the inhibitory activity of WT CTRP5 on ADIPOR1. Taken together, these data suggest that in healthy-iRPE cells, CTRP5 helps meet RPE energy demands by providing metabolic regulation to fine-tune ADIPOR1-dependent AMPK activity and this regulation is lacking in patient cells.

AMPK is a sensitive indicator of the cell energy status and is canonically activated by increased AMP:ATP ratio[40]. Therefore, we further characterized the AMPK activity (in serum-free media) of healthy and L-ORD-iRPE under conditions (increased AMP:ATP ratio) known to stimulate AMPK phosphorylation[41,42]. iRPE were incubated in serum-deprived media for 5 h to reduce intracellular ATP levels[36] followed by 30 min exposure to AICAR (an AMP analog) or BAM15 (a mitochondrial uncoupler that reduces ATP production) to further minimize the ATP cytoplasmic pool. Consistent with the data in Fig. 3a–c, increased cytoplasmic AMP (AICAR) or decreased ATP (BAM15) appropriately activated AMPK in healthy-iRPE, but not in L-ORD-iRPE (Fig. 3d, $p < 0.001$ and $p < 0.05$ for AICAR and BAM15, respectively). Taken together, these findings suggest that constitutively active ADIPOR1 in L-ORD-iRPE leads to chronic AMPK activation causing a failure in these cells to sense the AMP:ATP ratio making modulators of AMPK a potential target for therapeutic intervention.

Since AMPK binds to and activates PGC1α by direct phosphorylation we investigated whether L-ORD-iRPE exhibited changes in expression of PGC1α or its phospho-form. Although the levels of PGC1α were unchanged in L-ORD-iRPE compared to healthy (Fig. 3e, f, $p =$ ns), the levels of phospho-PGC1α at Ser-

570 that is known to inhibit its transcriptional activity[43] were increased 3.5-fold ($p < 0.05$) in L-ORD-iRPE compared to healthy-iRPE (Fig. 3g, h, $p < 0.05$), consistent with constitutively active AMPK and suppressed PGC1α activity in these cells.

To determine if constitutively active AMPK was indeed responsible for pathological features seen in L-ORD-iRPE, we subjected L-ORD-iRPE to adenine 9-beta-*d*-arabinofuranoside (1 μM, ara-A), a potent AMPK activity inhibitor[44]. Inhibition of AMPK activity in POS-fed (1 week) L-ORD-iRPE lessened sub-RPE APOE deposits, a key hallmark of L-ORD (Fig. 3i). Furthermore, AMPK inhibition led to a ~50% reduction in apical (Ap: $0.9 \pm 0.2$ ng/mL, $p < 0.05$) and basal (Ba: $0.3 \pm 0.04$ ng/mL, $p < 0.001$) VEGF secretion compared to untreated L-ORD-iRPE (Ap: $1.5 \pm 0.2$ ng/mL; Ba: $0.7 \pm 0.07$ ng/mL respectively, Fig. 3j). Taken together, these data suggest that the pathogenic *CTRP5* variant leads to pathological symptoms in L-ORD-iRPE cells likely due to constitutively activated AMPK signaling, thus incapacitating its ability to sense changes in cytosolic AMP:ATP ratio.

**Constitutively active AMPK disrupts PEDF-R mediated synthesis of retinal neuroprotectants and mitochondrial respiration in L-ORD-iRPE.** To determine the molecular mechanism driven by constitutively active AMPK pathology in L-ORD-iRPE cells, we examined the expression of its downstream signaling pathways. Pigment epithelium-derived factor (PEDF) and its receptor PEDF-R are particularly important targets of activated AMPK[45] because of their principal roles in regulating lipid metabolism and the release of free fatty acids. The expression (Fig. 4a) and apical localization (Fig. 4b) of PEDF-R (red) in L-ORD and healthy-iRPE were comparable, but we found a 50% reduction in apical/basal ratio of PEDF secretion in L-ORD-iRPE (L-ORD-iRPE: $5.5 \pm 1.1$; healthy-iRPE: $13.2 \pm 3.4$; Fig. 4c, $p < 0.05$). PEDF binding to PEDF-R stimulates the enzymatic phospholipase A2 activity of PEDF-R[46]. Accordingly, we observed a 44% reduction in phospholipase A2 activity of PEDF-R in L-ORD-iRPE (L-ORD-iRPE: $56.1\% \pm 7.1$; healthy-IRPE: $100\% \pm 16.9$; Fig. 4d, $p < 0.05$), consistent with the observed reduced apical secretion of PEDF.

To confirm that reduced phospholipase A2 activity was indeed associated with activated AMPK, we serum-starved healthy-iRPE cells to increase the levels of pAMPK. Following 24 h of serum starvation, phospholipase A2 activity was decreased by ~30%, suggesting that elevated pAMPK levels in L-ORD-iRPE are responsible for reduced phospholipase A2 activity in these cells (control: $100 \pm 10\%$; serum-starved: $72 \pm 7\%$; Fig. 4e, $p < 0.05$). Phospholipase A2 activity is required for mitochondrial function and the breakdown of phospholipids from phagocytosed POS into biologically active compounds such as DHA, eicosanoids, and Neuroprotectin D1 (NPD1)[26]. Reduced phospholipase A2 activity in L-ORD-iRPE resulted in 2.5-fold lower mitochondrial basal respiration rates (L-ORD-iRPE: $69.1 \pm 4.6$ pmol/min; healthy-iRPE: $171.1 \pm 7.0$ pmol/min; $p < 0.05$) and threefold lower ATP production ($21.4 \pm 2.2$ pmol/min) compared to healthy-iRPE ($58.7 \pm 16.1$ pmol/min, $p < 0.05$) (Fig. 4f, g). In addition, the NPD1 production was ~10-fold lower in L-ORD-iRPE ($1.1 \pm 0.4$ pg) compared to healthy-iRPE ($10.8 \pm 0.8$ pg; Fig. 4h). We found that L-ORD-iRPE phagocytose higher amounts of POS, as confirmed by a flow cytometry-based phagocytosis assay where ~50% more POS uptake was observed (Fig. 4i, $p < 0.01$). This likely occurred as a compensatory mechanism to offset lower NPD1 production by increasing the available free DHA pool-size[26]. But despite increased POS uptake, reduced phospholipase A2 activity suggested that L-ORD-iRPE have a lower ability to digest POS[47]. In agreement with this interpretation, the levels of

docosahexaenoic acid (DHA), a substrate from which very long-chain polyunsaturated fatty acids are derived, were found to be ~25% lower in apical media from L-ORD patient iRPE fed photoreceptor outer segments (POS) ($n = 12$ transwells per clone × 5 days; Supplementary Fig. 5c). Collectively, these results suggest that chronically active AMPK-perturbed PEDF/PEDF-R activity triggering pathological changes in L-ORD-iRPE via altered lipid metabolism, reduced ability to digest POS, reduced secretion of cytoprotective compounds needed by photoreceptors, and lowered mitochondrial respiration.

Recently, gene augmentation was shown to rescue disease phenotype in patient iPSC-derived RPE cells for a dominant maculopathy caused by mutations in gene BEST1[48]. We asked if a similar gene therapy approach was feasible for L-ORD-iRPE. Patient cells were transduced with progressively higher amounts of lentivirus constructs expressing GFP and WT CTRP5. GFP fluorescence confirmed increased expression both with increasing MOI and duration after transduction (Supplementary Fig. 6a, b). With increasing MOI of the WT CTRP5 expressing construct, 6–7x higher expression of CTRP5 on the apical side and 3–4x higher expression on the basal side of cells could be detected [Supplementary Fig. 6c, d; MOI 1.5: Ap ($p < 0.01$), Ba ($p < 0.001$); MOI 3.0: Ap ($p < 0.001$), Ba ($p < 0.05$)]. Consistent with increased secretion of CTRP5, pAMPK levels were reduced ~23% in L-ORD-iRPE with increasing concentration of WT CTRP5 (Supplementary Fig. 6e; MOI 0.5 and 1.5: $p = $ ns; MOI 3.0: $p < 0.05$). Again, consistent with the increased CTRP5 secretion and reduced pAMPK levels, mispolarized VEGF secretion was also corrected in WT CTRP5 overexpressed L-ORD-iRPE [Supplementary Fig. 6f; MOI 1.5: Ap ($p < 0.01$), Ba ($p < 0.001$); MOI 3.0: Ap ($p < 0.001$), Ba ($p < 0.05$)]. Together this data further reinforces our hypothesis that the dominant behavior of S163R mutation is through decreased secretion of CTRP5 in RPE cells and suggests that gene augmentation provides a potential treatment option for L-ORD patients.

**Metformin corrects L-ORD pathological phenotype.** To further determine if defective POS digestion and lipid metabolism contribute to the pathological features in L-ORD-iRPE cells, both healthy- and L-ORD-iRPE were fed photoreceptor outer segments (POS) for 7 consecutive days. POS-fed L-ORD-iRPE exhibited increased and variable cell size (mean area ± SD: $135.4 \pm 55.0$ μm$^2$), when compared to healthy-iRPE ($95.8 \pm 44.0$ μm$^2$, $p < 0.001$), perimeter, major axis, and minor axis (Figs. 5a, b, 1e and Supplementary Fig. 7a–c) or when compared to unfed L-ORD-iRPE cells ($106.9 \pm 68.0$ μm$^2$, $p < 0.001$) (Fig. 1e and Supplementary Fig. 7a). POS feeding for 7 days also increased the expression of dedifferentiation-related genes in L-ORD-iRPE (Fig. 5c). Compared to healthy-iRPE, L-ORD-iRPE upregulated 53/84 genes (>4-fold) associated with dedifferentiation (e.g., *ESR1, PDGFRB, TMEFF1, KRT14, PTK2, SOX10, GSK3B, TSPAN13, WNT5a, GNG11, ITGAV, SNAI3,* and *MMP9*) following 1 week of daily POS feeding, suggesting a possible mechanism of RPE atrophy and L-ORD disease phenotype worsening due to life-long POS uptake by diseased patient RPE cells.

Metformin, an anti-diabetic drug[49], has been used in the past for its lipid-lowering effects and for reversing epithelial dedifferentiation[50]. We asked whether metformin treatment of L-ORD-iRPE could reverse some of the pathophysiology induced by POS feeding. L-ORD and healthy-iRPE were treated daily with POS and 3 mM metformin for 1 week. Notably, metformin treatment of L-ORD-iRPE mitigated the POS-induced increase in cell size (Area: L-ORD untreated: 135.4 μm$^2$ ± 55 vs. L-ORD metformin: 117.9 μm$^2$ ± 34.1, $p < 0.0001$, compare Fig. 5b to d, 5e,

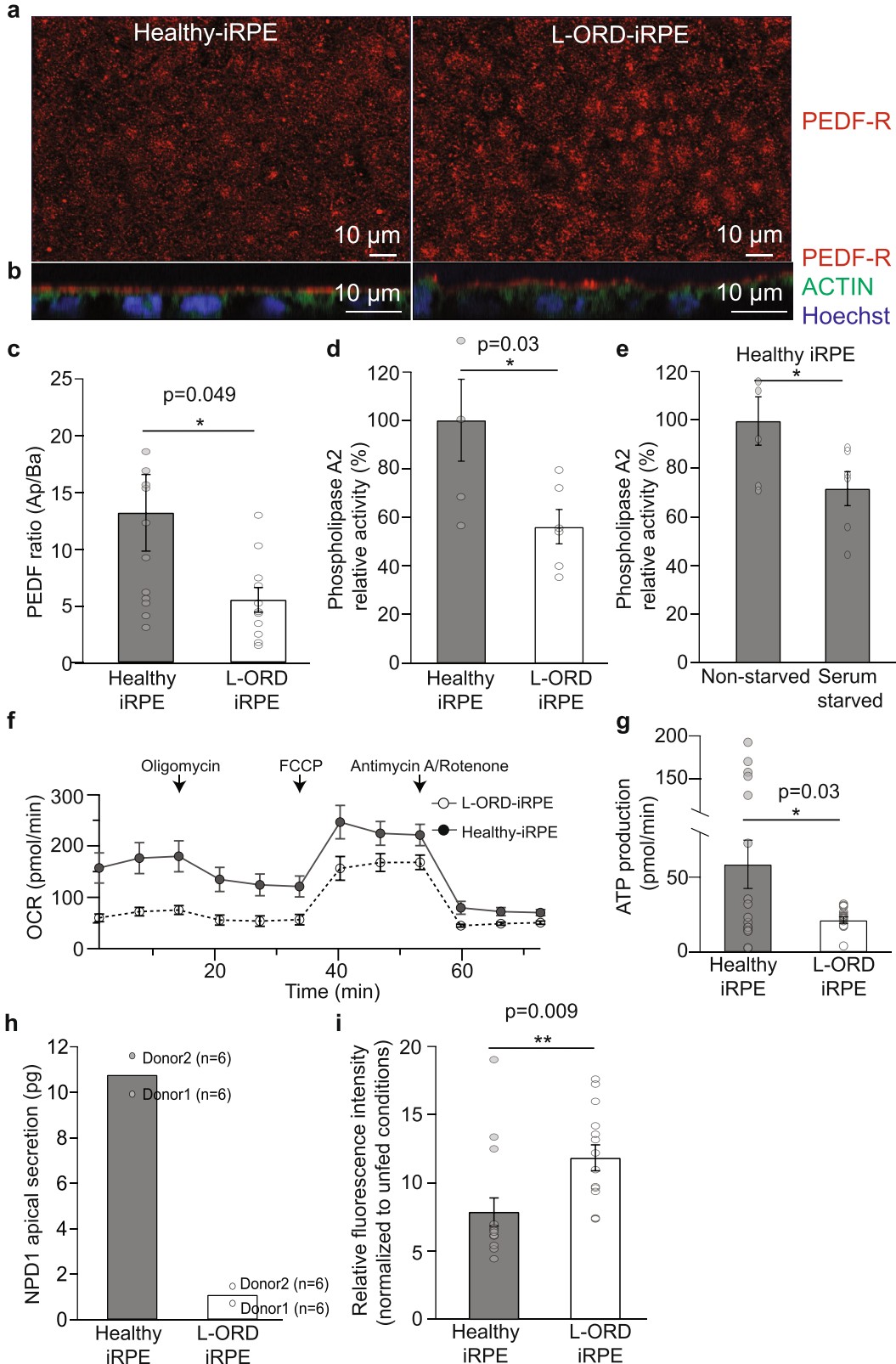

and Supplementary Fig. 8a). Additional cell morphometry metrics like perimeter, major axis, and minor axis of cells were also decreased upon metformin treatment of L-ORD-iRPE (Supplementary Fig. 8b–d). Metformin treatment (magenta) also downregulated the expression of 31 dedifferentiation genes by threefold or more (e.g., *ESR1, PDGFRB, TMEFF1, KRT14, PTK2, SOX10, GSK3B*, and *TSPAN13*) including 21/53 that were

overexpressed fourfold under POS feeding conditions (compare Fig. 5c, f, $p < 0.01$–$p < 0.001$).

Metabolic reprograming is proposed to play a critical role in RPE dedifferentiation[51]. To determine if metformin was able to restore fatty acid oxidative capacity through ketone body generation, we compared β-HB and NPD1 released apically in the presence of metformin. As seen in Fig. 5g, h, metformin

**Fig. 4 Altered lipid metabolism in L-ORD-iRPE contributes to reduced neuroprotective signaling. a**, **b** PEDF-R (red) immunolabeling in healthy-iRPE and L-ORD-iRPE. Cell nuclei (blue) and the actin cytoskeleton (phalloidin, green). Healthy-iRPE ($n = 8$), L-ORD-iRPE ($n = 6$). Scale bar: 10 μm. **c** Apical/basal ratio of PEDF secretion in L-ORD-iRPE ($n = 11$) as compared to healthy-iRPE ($n = 12$), measured by ELISA. **d** Phospholipase A2 activity in L-ORD-iRPE ($n = 6$) and healthy-iRPE ($n = 5$), measured by ELISA. **e** Phospholipase A2 activity of healthy-iRPE under basal and serum-starved (24 h) cell culture conditions ($n = 6$). **f** Seahorse assay results demonstrating oxygen consumption rate (OCR) in healthy ($n = 18$) and L-ORD-iRPE ($n = 18$) before and after the addition of mitochondrial respiration inhibitors (oligomycin, FCCP, antimycin A/rotenone). **g** ATP production measured from the Seahorse experiment in L-ORD-iRPE ($21.4 \pm 2.2$ pmol/min, $n = 18$) compared to healthy-iRPE ($58.7 \pm 16.1$ pmol/min, $n = 18$). **h** Apically secreted Neuroprotectin D1 (NPD1) measured by tandem mass spectrometry lipidomic analysis in POS-fed (4 h) L-ORD-iRPE ($n = 12$) and healthy-iRPE ($n = 12$). **i** Flow cytometry-based analysis of photoreceptor outer segment phagocytosis in L-ORD-iRPE and healthy-iRPE ($n = 14$). $*p < 0.05$, $**p < 0.01$.

addition increased β-HB and NPD1 release in L-ORD-iRPE ($p < 0.01$). This increased β-HB synthesis coincided with increased levels of *HMGCS2*, the enzyme catalyzing the committed step in ketone body generation, PEDF-R, and PRKAG1 an AMPK subunit (Supplementary Fig. 9a).

We then asked if metformin could rescue L-ORD-iRPE pathological phenotypes like sub-RPE APOE deposits and mispolarized VEGF secretion. Consistent with reduced phospholipase A2 activity and altered lipid metabolism in L-ORD-iRPE, a week of POS feeding exaggerated the sub-RPE APOE deposition. In images of POS-fed healthy-iRPE, APOE was found to be primarily on the apical side of cells as shown by their disparate integrated density values on the apical (yellow arrow) and basal (white arrow) sides (Fig. 6a —healthy-iRPE: top panel—low magnification and bottom panel—high magnification; quantification in Fig. 6e − $8.45 \pm 3.09$ a.u.). In contrast, POS-fed L-ORD-iRPE (Fig. 6b—L-ORD-iRPE: top panel—low magnification and bottom panel—high magnification) exhibited increased levels of APOE in the sub-RPE space (quantification in Fig. 6e; $46.38 \pm 2.51$ a.u.). The white arrow in Fig. 6b indicates a basal increase in APOE deposits in pores of the transwell membrane reported previously for cultured RPE cells with AMD phenotype[52]. Metformin treatment of L-ORD-iRPE ameliorated the accumulation of APOE-containing basal deposits (Fig. 6c—L-ORD-iRPE plus metformin: top panel—low magnification and bottom panel—high magnification; quantification in Fig. 6e; $13.6 \pm 4.6$ a.u., $p < 0.001$), but it didn't affect the overall APOE levels as confirmed by Western blot analysis (Fig. 6d).

Furthermore, metformin treatment corrected mispolarized VEGF secretion L-ORD-iRPE (compare Figs. 1i and 6f, L-ORD-iRPE, $n = 13$, Ap: $3.2 \pm 0.5$ ng/mL, Ba: $2.2 \pm 0.3$ ng/mL; metformin-treated L-ORD-iRPE, $n = 10$, Ap:$2.1 \pm 0.4$ ng/mL, Ba: $4.8 \pm 1.2$ ng/mL; Ap: $p = 0.1$; Ba: $p < 0.05$; healthy-iRPE, $n = 17$, Ap:$2.5 \pm 0.4$ ng/mL, Ba: $3.6 \pm 0.6$ ng/mL). Notably, the average VEGF Ba/Ap ratio calculated from individual transwells from L-ORD-iRPE ($0.9 \pm 0.1$) was significantly improved with metformin ($3.2 \pm 0.9$, $p < 0.01$) consistent with Ba/Ap ratios observed in healthy-iRPE ($1.8 \pm 0.2$, Fig. 1i). Figure 5g, h confirmed that metformin worked in L-ORD cells via improving mitochondrial activity as shown previously[53]. To check if this improvement also reset the AMPK activity in a feedback loop from mitochondria, we assessed the activity of the AMPK and its ability to sense cellular energy status defined by the AMP:ATP ratio. L-ORD-iRPE treated with metformin resulted in an increased ability of AMPK to respond to AICAR stimulation compared to L-ORD-iRPE not treated with metformin (Fig. 6g compare with Fig. 3d). This result suggests that metformin relieved chronic activation of AMPK and restored normal energy homeostasis. Taken together, these data indicate that L-ORD-iRPE are susceptible to lipid-stress induced dedifferentiation and that metformin can reactivate mitochondrial activity, reduce lipid stress, and resensitize AMPK, likely in a feedback loop. All together these changes in cellular metabolism alleviate pathological dedifferentiation of cells through downregulating dedifferentiation gene expression,

reducing sub-RPE APOE deposition, and correcting mispolarized VEGF secretion.

## Discussion

The monogenic dominant origin and pathological features of L-ORD have provided the basis for developing a patient-specific in vitro model improving our understanding of RPE pathogenesis that leads to vision loss in patients. Here, we reproduced three well-documented L-ORD disease cellular phenotypes—sub-RPE deposits, mispolarized VEGF secretion, and RPE atrophy[6] in patient RPE cells in vitro and provided direct evidence of the dominant-negative effect of the mutant CTRP5 protein. In human iPSC-derived RPE cells, we test the hypothesis that CTRP5 curtails ADIPOR1-driven AMPK signaling, which acts as a metabolic sensor to help control energy homeostasis, resistance to stress, and aging[54]. More specifically, chronic AMPK stimulation causes an imbalance in its ability to sense changes in intracellular AMP:ATP ratio, leading to diminished fatty acid utilization and oxidative phosphorylation, mispolarized VEGF secretion, the formation of sub-RPE deposits, and RPE atrophy.

Our analysis provides a molecular link between observations of sub-RPE deposits, RPE atrophy, and the pathogenic *CTRP5* variant that effectuates AMPK to become constitutively active resulting in diminished PEDF signaling in L-ORD-iRPE cells. Reduced phospholipase A2 activity of the PEDF receptor is known to reduce the efficiency of fatty acid metabolism, oxidative phosphorylation, and POS digestion[25,55]. This was confirmed in L-ORD-iRPE cells, which after a week of POS feeding exhibited decreased NPD1 and DHA secretion. Reduced phospholipase A2 activity and reduced PGC1α activity lower the ability of the mitochondria to generate ATP using oxidative phosphorylation. It has been reported that lower oxidative phosphorylation in RPE cells leads to increased glycolytic rate, which has been linked to cellular dedifferentiation in the RPE, as seen in our L-ORD cells[56,57]. Metabolic defects induced RPE dedifferentiation has been shown previously leading to photoreceptor degeneration in a mouse model[58]. Overall, our work provides a basis for understanding how the L-ORD-associated *CTRP5* variant disrupts lipid metabolism and DHA availability and leads to RPE atrophy and photoreceptor degeneration clinically observed in L-ORD patients.

The recycling of DHA through the intercellular matrix is a tightly regulated process involving interphotoreceptor retinoid-binding protein (IRBP)[59] that also traffics retinoids in the extracellular space between photoreceptors and the RPE. A similar dual role for ADIPOR1 has also been reported. In addition to stimulating AMPK activity, it also regulates DHA uptake in photoreceptors and the RPE[17]. Although DHA is enriched in photoreceptors and has long been shown to be integral for photoreceptor outer segment renewal[60], DHA is now appreciated as a precursor for many bioactive lipids, such as NPD1, that are important for maintaining the functional integrity of the photoreceptors[26,61].

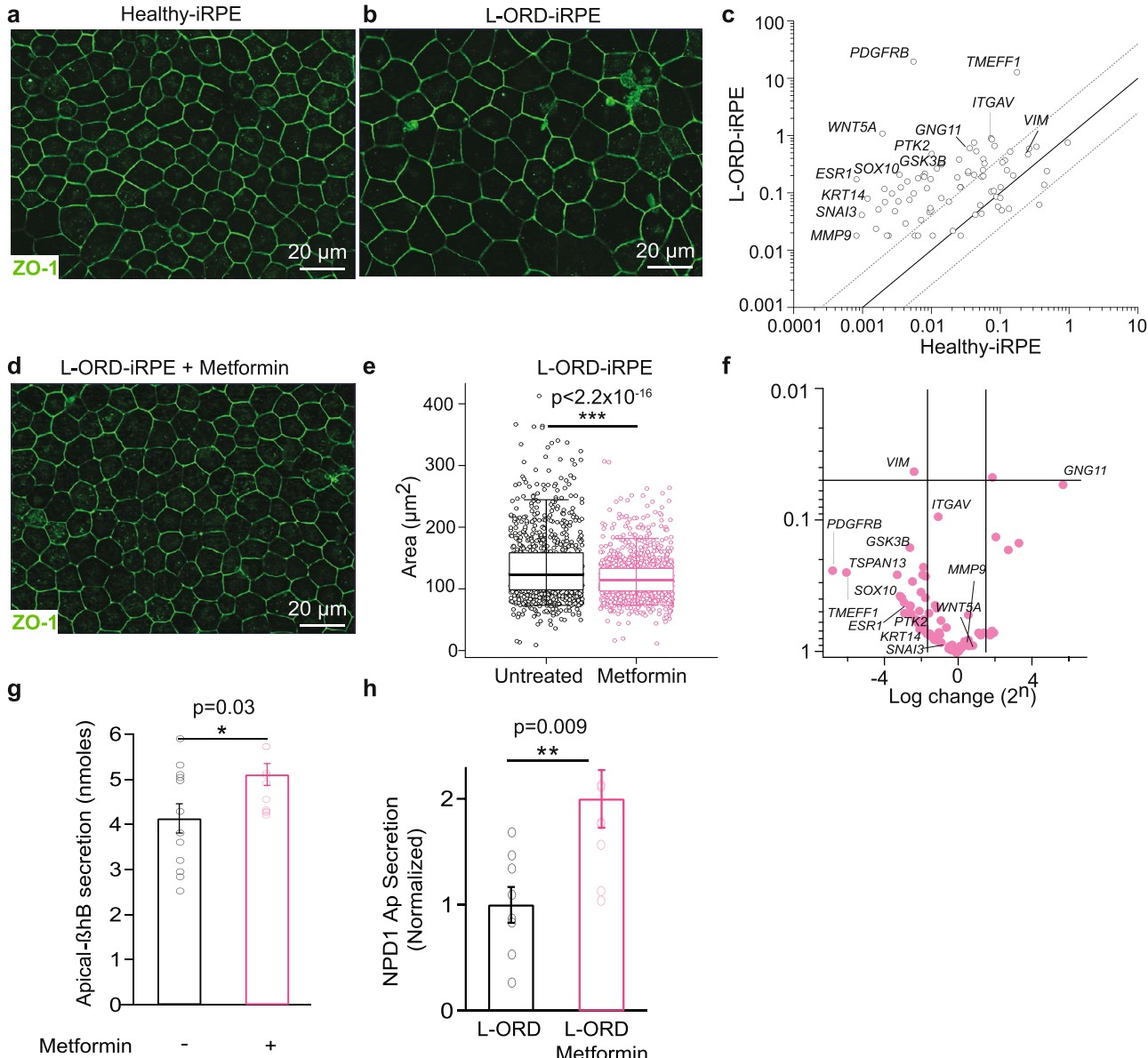

**Fig. 5 Metformin counteracts the increased susceptibility to dedifferentiation in L-ORD-iRPE. a, b** Representative immunofluorescent images of the membrane marker ZO-1 (green) in healthy (**a**) and L-ORD-iRPE (**b**) following 7 consecutive days of POS uptake. Scale bar 20 μm. **c** The effect of POS uptake on the expression of dedifferentiation-related genes in L-ORD-iRPE compared to healthy-iRPE. A dashed line indicates a fourfold difference. Housekeeping genes: *ACTB* and *GAPDH*. **d** Concurrent treatment of L-ORD-iRPE with metformin (3 mM) on POS-induced increase in cell size (ZO-1, green) after 7 days of POS uptake. Scale bar 20 μm. **e** Quantification of cell area after 7 days of POS uptake and metformin (3 mM) treatment in L-ORD-iRPE. Cells were labeled with anti-ZO-1 antibody and area was quantified using an AI-based algorithm[80], low whisker: 5% of data, low hinge: 25% of data, midline: median, high hinge: 75% of data, high whisker: 95% of data. ($n = 6$ images). **f** Expression of 31 dedifferentiation-related genes in metformin-treated (magenta) L-ORD-iRPE (fed POS for 7 days) compared to untreated cells. A dashed line indicates a fourfold difference. Housekeeping genes: *ACTB* and *GAPDH*. **g** Apically secreted beta-hydroxybutyrate (β-HB) in L-ORD-iRPE after 1 week of metformin treatment. Cells were supplied with a β-HB metabolic substrate, BSA-palmitate conjugate, for 3 h before measuring β-HB levels ($n = 12$). **h** Secreted NPD1 in untreated ($n = 8$) and metformin-treated L-ORD-iRPE ($n = 9$) measured by tandem mass spectrometry lipidomic analysis. Cells were POS-fed for 24 h prior to media collection. ∗$p < 0.05$, ∗∗$p < 0.01$, ∗∗∗$p < 0.001$.

The highest expression of CTRP5 protein is noted in the eye and adipose tissue[9,10]. To date, most work on the role of CTRP5 and the adiponectin pathway has been based on clinical or in vitro studies performed in non-eye tissues[62,63]. In myocytes, CTRP5 was shown to regulate the phosphorylation of AMPK, thereby regulating fatty acid oxidation in an autocrine manner[36]. Because of the structural homology between ADIPONECTIN and the CTRP family of proteins, ADIPOR1 has been proposed as a receptor for the CTRP family of proteins as well[16]. Consistently, our in silico modeling demonstrated that CTRP5 trimers could interact with the known ADIPONECTIN docking site on ADIPOR1. Furthermore, in RPE cells a strong co-labeling of CTRP5, specifically with ADIPOR1 and not ADIPOR2, was observed. Consistent with the immunolabeling data Co-IP confirmed physical interaction between ADIPOR1 and CTRP5. Our results support direct interaction between the two proteins—similar to CTRP9's interaction with ADIPOR1 in the brain[64].

Our data suggest that reduced secretion of CTRP5 is likely because oligomers of wildtype and mutant protein are trapped inside the cell. An increase in expression of the FLAG-tagged

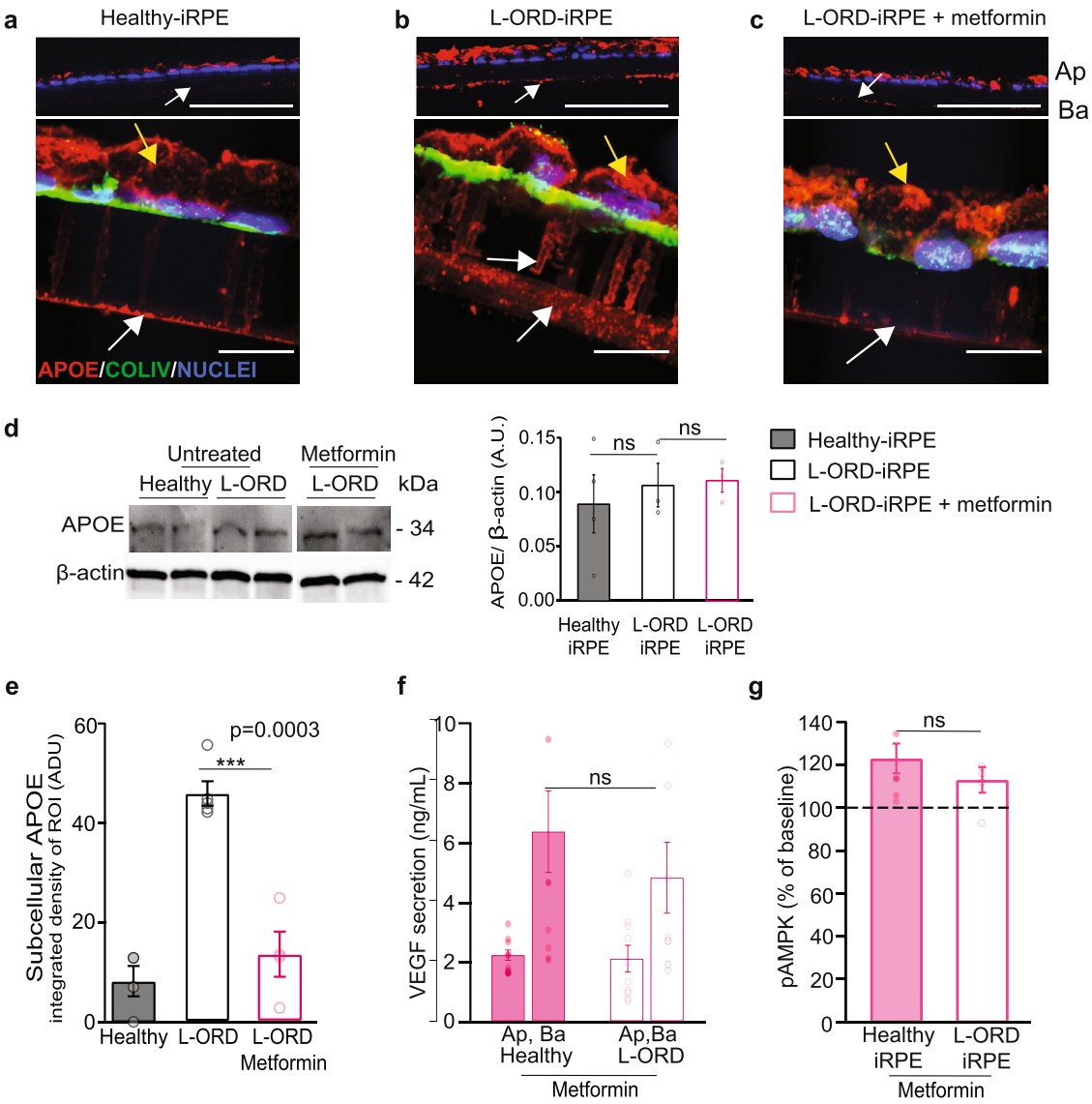

**Fig. 6 Metformin ameliorates L-ORD cellular phenotypes. a–c** Immunofluorescent images of cryosectioned healthy (**a**) and L-ORD-iRPE [without metformin (**b**), with metformin (**c**)] monolayers fed POS for 7 consecutive days and stained for APOE (red), COLLAGEN IV (COLIV, green), nuclei with Hoechst 33342 (blue). Top panels (**a–c**) show 20x magnification images, scale bar: 50 μm. Representative 60x magnification images are shown in bottom panels (**a–c**), scale bar: 10 μm. Apical APOE is indicated with yellow arrows and basal with white arrows ($n = 4$) **d** Representative Western blot and quantification of APOE signal in healthy ($n = 4$) and L-ORD-iRPE without ($n = 3$) and with metformin treatment ($n = 3$). **e** Quantification of APOE signal from images shown in panels **a–c**. ($n = 4$ for healthy-iRPE; $n = 5$ for L-ORD-iRPE; $n = 4$ for L-ORD-iRPE with metformin). **f** ELISA-based measurements of apical and basal VEGF secretion in metformin-treated healthy and L-ORD-iRPE ($n = 10$; Ap: 2.07 ± 0.4; Ba: 4.8 ± 1). **g** pAMPK levels in response to AICAR (2 kJ/mol) in L-ORD-iRPE ($n = 6$) treated with metformin, measured by ELISA.

mutant CTRP5 increased the intracellular signal for V5-tagged WT CTRP5 in a dose-dependent manner suggesting the mutant variant traps the WT inside the cell and lowers its secretion. The heterooligomers of WT-mutant CTRP5 are likely trapped in an endo-lysosomal compartment inside the cell and likely targeted for lysosomal degradation. Pearson's correlation coefficient clearly showed the strongest co-labeling of CTRP5 with LAMP1/2 and ATG5, especially when lysosomal degradation was blocked using bafilomycin A1. Our results differ from Shu et al. that showed endoplasmic reticulum localization of WT and mutant CTRP5[19]. This discrepancy is likely because Shu et al. used an overexpression system of either WT or mutant CTRP5. A similar discrepancy exists between our data and another study looking at overexpression of p.Ser163Arg CTRP5 in a mouse model[65]. Dinculescu observed large, round intracellular globular aggregates

of CTRP5 in this model[65], which are not seen in our L-ORD-iRPE. We are detecting endogenous expression of hetero-oligomers of WT CTRP5 as opposed to overexpression of WT or mutant proteins that was done in an RPE cell line or an adult mouse RPE.

All three reported variants (p.Gly216Cys, p.Pro188Thr, and p.Ser163Arg) of the CTRP5 globular domain are located on the inter-subunit interface with ADIPOR1. Consistently, our in silico modeling implies that all three pathogenic variants decrease the protein stability (>8 kJ/mol) and reduce the likelihood that the WT/mutant CTRP5 heterooligomers would interact with its receptor, ADIPOR1. Based on our data, we suggest that the dominant nature of L-ORD-associated variants is due to a combination of these two phenomena: (1) that heterooligomers of mutant and WT CTRP5 proteins are retained within the RPE;

and, (2) that there is a reduced affinity between secreted mutant/WT CTRP5 oligomers and ADIPOR1. Here we suggest that in healthy-RPE, native CTRP5 acts as an inhibitor for ADIPOR1 receptors since apical supplementation of gCTRP5 resulted in reduced pAMPK activity (Fig. 3b). In L-ORD, reduced availability of mutant/WT CTRP5 and reduced binding affinity of mutant/WT CTRP5 multimers leads to chronic activation of ADIPOR1. These results are further confirmed by the significantly higher constitutive AMPK phosphorylation seen in L-ORD-iRPE cells and by the absent effect on ADIPOR1 ceramidase activity (Supplementary Fig. 4c, d). The use of a gene augmentation approach to overexpress WT CTRP5 in L-ORD-iRPE rescued disease phenotypes—it increased the secretion of CTRP5, reduced pAMPK levels, and corrected mispolarized VEGF secretion. This data further supports our results that entrapment of the WT-variant by the mutant variant inside the cell is the likely reason for the dominant disease phenotype and it provides a gene therapy approach to treat L-ORD patients.

Taken together, this evidence suggests a role for CTRP5-ADIPOR1 interactions in mediating the disease phenotype in L-ORD patient cells. The loss of CTRP5 inhibitory actions on ADIPOR1 signaling results in constitutively active AMPK—a master regulator of lipid metabolism. This is a critical outcome for RPE, since it ingests a high-fat diet of photoreceptor outer segments on a daily basis[66]. Similar to our findings, AMPK inhibition of PEDF expression has been reported in adipocytes and hepatocytes[67]. PEDF is known to prevent intracellular calcium overload in rd1 mouse photoreceptors by acting upon plasma membrane $Ca^{2+}$ ATPase pumps[68]. Additionally, it suppresses RPE dedifferentiation and migration and improves RPE mitochondrial function[69]. In L-ORD-iRPE, chronic AMPK activation reduces PEDF expression—thereby significantly reducing the phospholipase activity associated with PEDF-R, which is required to mobilize free fatty acids from digested POS.

Metformin is a widely prescribed drug for treating type 2 diabetes and acts by lowering blood glucose among patients by reducing hepatic glucose production, but its mechanisms of action are broad[70]. It's been proposed that metformin may indirectly influence AMPK activity via improvement in mitochondrial oxidative phosphorylation, fatty acid oxidation, protein translation, or by altering AMPK binding/interaction with LKB1[71–74]. Through improved ATP production via mitochondria, metformin likely reduces cellular dependency on glycolysis[75] and reverses RPE dedifferentiation[50]. The same phenomenon was observed in metformin-treated L-ORD-iRPE cells. Importantly, metformin has also been shown to delay retinal degeneration in RD10 mice and to protect the RPE from sodium iodate-induced damage[76]. Of notable interest in L-ORD-iRPE is the increased expression of: (1) PEDF-R, which stimulates the release of free fatty acids; (2) HMGCS2, which catalyzes ketogenesis—converting excess products of fatty acid oxidation (acetyl-CoA) to β-hydroxybutyrate, a metabolic substrate utilized by the retina[23]; and (3) PRKAG1, the AMP/ATP binding subunit of AMPK which has been shown to influence the degree of AMPK stimulation. Combined, these three effects of metformin remodel lipid metabolism in the RPE, improve fatty acid utilization and alter APOE distribution thus reducing sub-RPE lipid deposition.

### Implications of clinical and in vitro data.
The RPE is situated between the neurosensory retina and the vascular choroid and serves as a metabolic gatekeeper achieving homeostasis by balancing the energy demands of the retina while maintaining the integrity of the blood–retina-barrier. Interestingly, under normal circumstances, the dominant secretion of VEGF by the RPE is basal toward the choroid[77]. In L-ORD however, the underlying inherited metabolic defect alters this secretion profile and is reversed—favoring secretion toward the distal retina. This reversal is corrected by treatment with metformin and VEGF can return to protecting the choroid. Hence future investigations are needed to further evaluate the pathophysiological aspects of this relationship.

Metformin and other AMPK regulators are in clinical use[78]. Thus, the connection with altered AMPK activity provides a potential treatment opportunity for L-ORD and for other retinal degenerative diseases. Our in vitro data indicates that the disease-associated phenotypes of L-ORD were strongly influenced by early intervention/pretreatment with metformin. As is the case in other epithelia where epidemiological evidence suggests that metabolic imbalance contributes to disease onset and/or progression in diabetes and cancer[79], the therapeutic mechanism of metformin is likely through a shift in the metabolic state toward the norm and prevention of dedifferentiation. Overall, our study provides novel insights into the role of CTRP5 in the RPE, how the pathogenic variant in L-ORD causes dominant disease, and provides further evidence that metformin can be a beneficial intervention for the treatment of L-ORD.

## Methods

**Human iPSC generation using patient-derived fibroblast cells**. Written approvals for human skin-tissue collection and iPSC generation were obtained as part of an NEI IRB-approved protocol, 11-E1-0245. Written informed consent was received from participants prior to inclusion in the study. The skin biopsy samples were "coded" where no personally identifiable information (PII) was directly available to the researchers. Human biospecimens were stored according to the Guidelines for human biospecimen storage, tracking, sharing, and disposal within the NIH Intramural Research Program.

Skin punch biopsies were obtained from a clinically and genotypically confirmed family of L-ORD patients (one female age 57, one male age 51) and unaffected siblings (two males ages 58 and 54) (NIH Clinical Center) with their signed consent. Affected members of this family were diagnosed with late-onset retinal degeneration contributing to symptomatic night blindness and progressive loss of peripheral vision. Affected individuals had clinical molecular genetic testing confirming that they carry the familial p.Ser163Arg missense variant in CTRP5. (See Supplementary Methods for iPSC generation, validation, and differentiation into iRPE).

**Study design**. In this study, we generated iPSC-RPE from patients with late-onset retinal degeneration (L-ORD) and their unaffected ("healthy") siblings to use as an in vitro model to investigate L-ORD pathogenesis. By including multiple donors from the same family (~50% of the genome shared) and generating multiple iPSC clones from each donor we mitigated the likelihood that phenotypic differences could be attributed to differences in genetic background. Additionally, the cellular sources for all the iPSCs were the same: skin fibroblasts and all iPSCs were derived using a non-integrating approach (Sendai virus). The passage numbers and time in culture were comparable across the entire study. iPSC-RPE were seeded onto transwells at 250k/well and matured for >6 weeks before being used for the following assays to determine the underlying disease mechanism and to consequently design a method to rescue the disease phenotype in vitro. Specifically, we investigated how the single missense mutation in CTRP5 affects its expression, secretion, and its interaction with other proteins. We identified ADIPOR1 as a likely candidate for interaction and confirmed this by co-immunoprecipitation. Since L-ORD is a dominant monogenic disorder, it was presumed that the effect of the mutation would be highly penetrant which was confirmed by the significant reduction in CTRP5 secretion in all L-ORD iPSC-RPE. Thus, the downstream effects of this phenotypic change (i.e., AMPK activity) outweighed any confounding factors that may be present in this model. All subsequent experiments and assays are described in detail below and in the Supplementary Methods.

**RNA extraction and cDNA synthesis**. iRPE on transwells were washed 3x with DPBS. RNA was extracted using NucleoSpin RNA (Macherey-Nagel, Düren, Germany, #740955.50) per manufacturer's instructions. Briefly, cells were lysed by placing them in RA1 Lysis buffer 350 μL + 3.5 μL β-mercaptoethanol/well (Thermo Fisher Scientific, #21985023) for 10–15 min prior to pipetting the cell lysate and immediately flash freezing with liquid nitrogen. Elution of RNA was performed, and cDNA was prepared from mRNA using SuperScript III First-Strand Synthesis kit (Thermo Fisher Scientific, #11904-018). cDNA was diluted to 1 ng/μl.

**qPCR**. Using validated primer sets (BioRad Laboratories, Inc., Hercules, CA) we ordered a custom-designed 96 gene PrimePCR four-quadrant, 384 well plates for

(1) human AMPK and fatty acid metabolism (Biorad, #10025214), and a 384 well plate for human genes related to (2) epithelial-to-mesenchymal transition (EMT) (Biorad, #10034487). Quantitative polymerase chain reaction (qPCR) was run in duplicate or triplicate using RT$^2$ SYBR Green qPCR Mastermix (Qiagen, Hilden, Germany, #330503) on a ViiA 7 Real-Time PCR System (Thermo Fisher Scientific, #4453536). Housekeeping gene (*HPRT* and *TBP*) transcript levels were used as internal controls. Genes with non-detect Ct values in both unaffected siblings and L-ORD patients were removed from the analysis. Relative quantification was calculated using the $2^{-\Delta\Delta CT}$ method. For qPCR primer analysis of individual genes of interest (*CTRP5*, *MFRP*)/housekeeping gene (*RPL13A*) primers were ordered from Integrated DNA Technologies (Coralville, IA). The following primers and primer sequences were used.

**CTRP5**. FWD: GCAAGTTCACCTGCCAGGTGCC
REV: GGATTCGCCATTCTTCACCAGATC

**MFRP**. FWD: TCACCAACTGCTCTGCACCTGG
REV: AGTCAAACTTGCACTCGTCCTGAG

**Transmission electron microscopy**. Samples for TEM were processed as previously described (Ogilvy et al., 2014). Briefly, iRPE were washed 3x in PBS and then fixed in PBS-buffered glutaraldehyde (2.5% at pH 7.4) (VWR, Randor, PA, #102092-014) and treated with 0.5% ice-cold PBS-buffered osmium tetroxide (Electron Microscope Sciences, #19190) and embedded in epoxy resin. Ultrathin sections (~90 nm thick) were cut on an ultramicrotome and mounted on 200-hexamesh copper grids (Electron Microscopy Sciences, #G200H-Cu). TEM images were acquired on a JEOL JEM-1010 transmission electron microscope (JEOL, Peabody, MA).

**Scanning electron microscopy**. iRPE on transwells were fixed overnight in 2.5% glutaraldehyde + 1% formaldehyde in 0.1 M sodium cacodylate buffered solution. Samples were cut to fit the sample holder and underwent ethanol dehydration progressing sequentially through ddH20 (5 min), 30% ethanol (10 min), 50% ethanol (10 min), 70% ethanol (20 min), 90% ethanol (5 min), 95% ethanol (5 min), 100% ethanol (5 min), 100% ethanol (5 min), 100% ethanol (5–20 min) (Sigma, #459844500 ML) while transported to a critical point drying (CPD) machine (Leica, Wetzlar, Germany, #EM CPD300) where they were further dehydrated. Samples were then mounted onto SEM pin stub mounts (Ted Pella, Redding, CA, #16111-9) and coated with 10–15 nm of gold using a low vacuum sputter coater (Leica, #EM ACE 200) and stored in a desiccator until imaged on a Zeiss EVO MA 10 scanning electron microscope (SEM).

**Transepithelial electrical resistance (TER)**. Resistance measurements were taken using the STX2 electrode set with the EVOM2 meter (World Precision Instruments, Sarasota, FL). The unit area resistance was calculated by multiplying the measured resistance by the effective membrane area (1.12 cm$^2$).

**Immunostaining of iRPE monolayers**. iRPE monolayers were fixed in 4% PFA (Electron Microscopy Sciences, #15710) for 10 min, washed 3x in PBST (1x PBS, 0.5% Tween20), and permeabilized in ICC blocking buffer (1x PBS, 1% BSA, 0.25% Tween20, 0.25% Triton X-100) for 1 h. iRPE were incubated overnight at room temperature with antibodies diluted in ICC blocking buffer at the following dilutions: CTRP5 (1:100, Bioss, #bs-11717R), MFRP (1:100, R&D Systems, #AF1915), APOE (1:100, EMD Millipore, #AB947), Collagen IV (1:100, EMD Millipore, #ab756p), Ceramide (1:100, Enzo, #ALX-804-196-T050), ADIPOR1 (1:300, Enzo, #ALX-210-645-C200), ADIPOR2 (1:250, EMD Millipore, #MABS1166), EZRIN (1:200, Sigma, #E8897), ZO-1/TJP1 (1:100, LS-Bio, #LS-B9774), EEA1 (1:100, Abcam, #ab70521), (1:500, LAMP1, Abcam, #ab25245), (1:500, LAMP2, Abcam, #ab25631), (1:200, ATG5, Thermo Fisher Scientific, #MA5-38452). For co-localization studies of CTRP5 with endo-lysosomal markers, iRPE were treated with and without bafilomycin (50 nM, Sigma, #B1793-10UG) for 3 h prior to fixation.

Cells were washed 3x in ICC buffer and secondary antibodies were added at 1:1000 dilution. The following conjugated antibody was used to stain cell borders: ZO-1 conjugate Alexa Fluor 488 (1:100, Thermo Fisher Scientific, #339188). Samples were washed 3x in ICC blocking buffer and mounted onto glass slides using Fluoromount-G aqueous mounting medium (Southern Biotech, Birmingham, AL, #0100-01). Images were acquired on a Zeiss Axio Imager M2 inverted fluorescent microscope with Apotome 2 and Zen 2012 software or a Zeiss 800 confocal microscope (Carl Zeiss AG, Oberkochen, DE). Airyscan images were taken on a Zeiss LSM 880 with Airyscan. All similarly stained samples were imaged under the same exposure times and adjusted to the same contrast settings before being exported. For staining of APOE on transwells and cryosectioning of APOE-stained iRPE see SI Methods.

**Shapemetric analysis**. Morphometric analysis of RPE cells was performed using REShAPE, an open-access cloud computing-based automated image analysis platform. RPE monolayers were stained for ADIPOR1 or ZO-1 to identify cell

borders through image segmentation. Analysis was performed on 20x or 40x images rescaled to 20x. Shapemetric values are displayed as Mean ± SD (See SI Methods for further details).

**VEGF Elisa of cell culture supernatants (R&D systems, DVE00)**. Prior to the experiment, selected transwells were washed 1x with DPBS (Gibco, #14190-144) and media was replaced with fresh 5% serum-containing RPE media. In select experiments, 1 μM 9-beta-*d*-arabinofuranoside (ara-A) (Sigma, #A5762-1G), an AMPK inhibitor, was added to the media. Following media change, the transwell plate was placed in a CO2 incubator for 6 h. Following incubation, the media from the apical and basal sides of the transwells were collected on ice. The conditioned media was centrifuged at 13,200 rpm at 4 °C for 15 min. The supernatant was transferred to fresh 1.5 mL Eppendorf tubes and frozen on dry ice and stored at −80 °C. The assay was completed according to the manufacturer's instructions. (Note: samples were diluted 20-fold in diluent buffer).

**CTRP5 Elisa**. iRPE were incubated for 24 h in phenol-red/serum-free RPE media. Apical and basal media was collected on ice and centrifuged at 13,200 rpm for 15 min at 4 °C. The supernatant was frozen on dry ice and stored at −80 °C. CTRP5 was measured using a CTRP5 Elisa kit (Aviscera Bioscience, Santa Clara, CA, Cat#: SK00594-06) per the manufacturer's instructions. Apical media was diluted by 16-fold, and basal media was diluted by 8-fold.

**Plasma membrane preparation and co-immunoprecipitation (CO-IP)**. Plasma membrane preparation of iRPE were prepared using a Plasma Membrane Protein Extraction Kit (Abcam, #ab65400) following the manufacturer's recommendations. In brief, 9 million iRPE cells were collected from transwells on ice. Cell lysates were prepared by homogenizing cells 30–50 times in premade Homogenization Buffer Mix in an ice-cold Dounce Homogenizer. Two passes are required to ensure thorough homogenization. Nuclei were pelleted and discarded, and the supernatant were further centrifuged at maximal speeds in a microcentrifuge and processed further to obtain total cell membrane and plasma membrane preps.

Immunoprecipitation was carried out using Dynabeads$^{TM}$ protein G Immunoprecipitation kit Thermo Fisher,#10007D per manufacturer's protocol. In brief, plasma membrane preps (obtained from 9 million cells) were resuspended in 300 ul of TBST (PBS + 0.5% Triton) and divided into three equal parts. Each part was probed with anti-AdipoR1 (Santacruz (D-9): #sc-518030, 10ug), PBST, or matching isotype(IgG), respectively. The latter two samples were control samples that were used to evaluate the nonspecific binding of CTRP5. CTRP5 in pull-down lysates were detected by Western Blot with an anti-CTRP5 antibody (same as previously reported).

**Generation of tagged CTRP5 wildtype and S163R mutant overexpression system**. Coding sequences of the Human CTRP5 wildtype or mutant variant S163R were cloned into a lentiviral vector, with a V5 or Flag tag included to its C-terminus, respectively. The lenti-construct was designed to express GFP or mCherry via IRES (Internal ribosome entry site) for monitoring of transduction efficiency and protein expression.

**Immunostaining of iRPE overexpressing tagged WT or MUT CTRP5**. iRPE monolayers derived from L-ORD patients were transduced with lentivirus containing tagged WT (V5)and MUT CTRP5(Flag), simultaneously using polybrene (6 ug/ml) containing media for 3 days. WT CTRP5 contains a V5 tag and MUT CTRP5 contains a Flag-Tag, both included on the C-terminus of the protein. Tagged WT CTRP5 was expressed at a constant level (MOI 0.5) and varying tagged S163R MUT CTRP5 (0.5, 1.5, 3). The cells were fixed in 4% PFA and permeabilized in ICC blocking buffer (1x PBS, 1% BSA, 0.25% Tween20, 0.25% Triton X-100) for 1 h. The iRPE were incubated with antibodies diluted in ICC blocking buffer at the following dilutions: V5 (1:250, Abcam, ab27671), Flag (1:200, Thermo Fisher, INC, #701629).

**Native immunogold labeling**. Primary antibodies for ADIPOR1 (1:300, Enzo, #ALX-210-645-C200) and CTRP5 (1:100, Bioss, #bs-11717R) were incubated for 1.5 h at 37 °C in 5% serum-containing RPE media. Cells were subsequently washed four times in 1x PBS (slowly with a pipette to protect the apical processes). The immunogold secondary antibodies 6 nm conjugated gold (Jackson Immunoresearch, #705-195-147) and 12 nm conjugated gold (Jackson Immunoresearch, #711-205-152) were added together each at a dilution of 1:500 for 1.5 h at 37 °C. Cells were washed four times in 1x PBS and fixed in 2.5% glutaraldehyde + 2.5% formaldehyde in 0.1 M sodium cacodylate buffered solution (Electron Microscopy Sciences, #15949) for 72 h and provided to the NEI histopathology core facility for preparation for TEM imaging.

**Phospho-AMPK assay**. Phospho-AMPK was measured using the AMPK [pT172] Elisa Kit (Thermo Fisher Scientific, #KHO0651) per the manufacturer's instructions. Cells were either collected at baseline at 37 °C in 5% serum-containing media, or under serum starvation (5 h) followed by 30 min treatment of 2 mM AICAR (Sigma, #A9978-5mg) or 100 nM BAM15 (Sigma, #SML1760-5mg). Cells

pretreated with metformin (3 mM) for 1 week were similarly tested under serum starvation (5 h) followed by 30 min of 2 mM AICAR.

In experiments to determine the effects of recombinant CTRP5 globular domain (gCTRP5, Adipogen, #AG-25A-0096-C100) or recombinant CTRP5 full length (Adipogen, #AG-40A-0142-C050) on pAMPK activation, samples were serum-starved for 5 h followed by 30 min incubation with recombinant CTRP5 reconstituted in water (gCTRP5: 0.2 µg/mL; full-length CTRP5: 0.2, 2, and 25 µg/mL). Cells were washed twice with cold 1x PBS prior to adding 200 µL of cell extraction buffer (Thermo Fisher Scientific, #FNN0011) and 2 µL Halt Protease Inhibitor cocktail (100x) (Thermo Fisher Scientific, #P-2714/78430) per well for ~5 min on ice. Samples were detached by pipetting and vortex three times at 10-min intervals. The extract was placed in microcentrifuge tubes and centrifuge at 13,200 rpm for 10 min at 4 °C. The clear lysate (supernatant) was collected and frozen on dry ice and stored at −80 °C. For the assay, the cell extract samples were diluted at 1:2.2.

**Western blot of iRPE cell lysates**. iRPE cells grown on transwells were lysed in RIPA buffer supplemented with Halt Protease Phosphatase Inhibitor Cocktail (Thermo Fisher Scientific, #78440). Total protein was quantified using BCA analysis (Thermo Fisher Scientific, #23227) per the manufacturer's instructions. Buffers for Western blotting: Running Buffer consisted of double deionized water, 10x Tris/Glycine/SDS (Biorad, #1610732). Transfer buffer consisted of double deionized water, 5x transfer buffer, and 20% ethanol (200 proof). Cell lysates were separated by denaturing gel electrophoresis and electroblotted onto a PVDF membrane. Blocking of nonspecific binding was sufficiently obtained with 5% BSA in PBST. Membranes were incubated 12–18 h at 4 °C with primary antibodies against PGC1α (Abcam, Cat#: ab54481) and Phospho-PGC1α (S571) (R&D Systems, Cat#: AF6650) and β-Actin (Cell Signaling, #4970 S). After 3x washes in PBST, the secondary antibodies were added (IR dye −800 and −680) and incubated for 1 h at room temperature. Western blots were imaged using the ChemiDoc MP Imaging System (Biorad).

**Phospholipase A2 assay**. Phospholipase A2 enzyme activities was measured using the EnzChek Phospholipase A2 Assay Kit (Invitrogen, Carlsbad, CA, #E10217) per the manufacturer's instructions. Baseline phospholipase A2 (PEDF-R) activity was measured from healthy and L-ORD-iRPE maintained at 37 °C in 5% serum-containing RPE media. Phospholipase A2 activity in response to elevated AMPK was measured after 24 h exposure to 0% serum-containing media in healthy-iRPE. Samples were collected in a native gel lysis buffer (150 mM NaCl, 1% Triton X-100, 1 M Tris-HCL). Samples were centrifuged at 13,200 rpm for 10 min at 4 °C. The supernatant was transferred to new tubes and frozen on dry ice and stored at −80 °C. A total volume of 100 µL was used following the standard assay protocol: 50 µL sample + 50 µL substrate-liposome mix.

**Cellular energetics (Seahorse)**. The Seahorse XFe96 Analyzer (Agilent, Santa Clara, CA) was used to measure oxygen consumption rate (OCR) and extracellular acidification rate (ECAR). Cells were plated at a density of 1000 cells per well and allowed to mature to confluency ~7 weeks. All four donors (two healthy, two L-ORD) were seeded onto the same plate (six replicate wells of each line were used and the responses across donors (healthy vs L-ORD) were averaged.

**LC-MS/MS**. We used a Xevo TQ-S mass spectrometer equipped with Acquity I Class UPLC (Waters Corporation, Milford, MA). CORTECS C18 column (2.7 µm particle size, 4.6 mm × 100 mm i.d.) (Waters Corporation, #186007377) for the separation of fatty acids and their derivatives. LC conditions are as follow; 45% of solvent A (H$_2$O + 0.01% acetic acid) and 55% of solvent B (MeOH + 0.01% acetic acid) with a flow rate of 0.6 mL/min was initially set at 0 min., then gradient was set for 85% B at 10 min, 98% B at 18 min, 100% B at 20 min, back to 55% B at 30 min. The capillary voltage was −2.5 kV, desolvation temperature was 600 °C, desolvation gas flow was 1100 L/h, cone gas was 150 L/h, and nebulizer pressure was 7.0 bar with a source temperature of 150 °C. MassLynx ver. 4.1 software (Waters Corporation) was used for the operation and recording of the data. Data were expressed as the change in the relative abundance of pro-homeostatic lipid mediators normalized with the internal standards.

**Photoreceptor outer segment (POS) phagocytosis assay**. The phagocytosis assay for assessing the functionality of iRPE was performed as mentioned previously (May-Simera et al., 2018; Sharma et al., 2019). Briefly, bovine outer segments were procured commercially (InVision Bioresources, # 98740) and labeled with pH-sensitive dye pHrodo red dye conjugates (Thermo Fisher Scientific, #P36600), as per the manufacturer's instructions. iRPE cells were fed with pHrodo labeled photoreceptor outer segment (POS) 10 POS/iRPE cells, for 4 h. Untreated cells were used as control. After 4 h, cells were washed four times with 1x PBS (Thermo Fisher Scientific, #10010-023), and incubated in 0.25% Trypsin-EDTA (Thermo Fisher Scientific, #25200056) for 20 min. The Trypsin-EDTA was aspirated off and replaced with RPE maintenance media. Cells were collected in a tube and centrifuged at 1200 rpm for 5 min. Cell pellets were washed three times with 1x PBS. After the final wash, pellets were resuspended in 600 µL of 0.1% BSA in 1x PBS; cell suspension was strained through a 40-micron cell strainer (BD

Biosciences, #14-959-49 A). About 10 µL of DAPI (5 µg/mL) was added to each tube that contained cell suspension. pHrodo signal (mean fluorescence intensity) from phagocytosed POS was measured using a MACSQuant Analyzer (Miltenyi Biotech) for a total of 10,000 counts/treatment. Data were analyzed by FlowJo software (BD Biosciences). The ratio of fed to unfed samples was plotted.

**β-hydroxybutyrate assay**. Ringer solution and BSA conjugated Na palmitate were prepared as described previously[24]. On the day of the experiment, 2 mM BSA-palmitate conjugate was added to Ringer's solution at 1:10 dilution. Ringer's solution containing (BSA-palmitate conjugate) was added to the apical chamber of iRPE cells grown on 12-well transwell filters. Ringer's solution (115 µl) was collected from the apical and basal chamber and analyzed for β-hydroxybutyrate (β-HB) at the 3-h time point essentially as described[23]. The β-HB levels were determined using the β-hydroxybutyrate LiquiColor kit (Stanbio, Boerne, TX, #2440-058). In this assay, reagent A was mixed with reagent B at a 6:1 ratio, and 150 µl of this mixture was added to 100 µl of samples or βHB standards in each well. The plate was light-protected and incubated at 37 °C for 1 h with gentle shaking before absorbance at 492 nm was measured.

**Statistics and reproducibility**. Paired sample (two-sided) t-test analysis was used to determine if the differences observed between healthy and L-ORD-iRPE were statistically significant ($P ≤ 0.05$). Statistical analysis was performed in MatLab and Microsoft Excel. The data were represented as Mean ± Standard Error unless otherwise noted (e.g., boxplots). Boxplots were performed in R version 3.6.2 or Igor Pro version 6.37. The 25th, median, and 75th percentiles correspond to the bottom, middle, and top of each box.

Co-localization analysis was performed using Pearson's correlation coefficient calculated by NIS Elements AR software (Nikon). Each experiment presented here uses averaged data from iRPE derived from four iPSC clones of two unaffected siblings (healthy-iRPE) and four iPSC clones of two patients (L-ORD-iRPE). For each figure, the number of replicates is indicated in the corresponding legend.

**Reporting Summary**. Further information on research design is available in the Nature Research Reporting Summary linked to this article.

## Data availability
All data, reagents, and materials are available upon reasonable request from the corresponding author (kapil.bharti@nih.gov). Source data for all figures are provided in Supplementary Data.

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

## Acknowledgements
We are grateful to the patients and family members that participated in this study. The authors are pleased to thank Robert Fariss, NEI Biological Imaging Core and Jizhong Zou (NHLBI iPSC core), Omar Memon, Jason Silver, and Vaisakh Rajan for technical assistance. The authors thank Dr. Brian Brooks and Dr. Rajendra Apte for helpful tips and comments. This work was supported by funds from the NEI Intramural Research Program to Dr. Kapil Bharti and Dr. Sheldon Miller, grant EY005121 and the Eye Ear Nose Throat (EENT) Foundation of New Orleans for Dr. Nicolas G. Bazan, grant EY026525 for Dr. Kathleen Boesze-Battaglia.

## Author contributions
Conceptualization, K.B.; Methodology, K.B., K.J.M. and R.S.; Investigation, K.J.M., R.S., M.N., K.C.G, Z.Q., D.O., D.B., M.F., C.Z., A.F., Y.V.S., M.A.-A., B.J., K.V.D., M.-A.K.G., J.C., A.G., B.G., Q.W., R.C.S., C.C., P.A.S. and R.B.H.; Formal analysis, K.J.M., R.S., M.N., D.O., N.G.B. and K.B.-B.; Project administration, K.J.M. and R.S.; Writing—original draft, K.J.M.; Writing—review and editing, K.J.M., R.S. and K.B.; Funding acquisition, K.B.; Resources, S.M. and K.B.; Supervision, K.B.

## Funding

## Competing interests
The authors declare no competing interests.
