## [Transparent Peer Review File · Communications Biology]

Reviewers' comments:

Reviewer #1 (Remarks to the Author):

Review of "AMPK modulation ameliorates dominant disease phenotypes of CTRP5 variant in retinal degeneration".

This is a very well-written and well-thought out and experimentally-executed manuscript demonstrating disease modelling of sibling patient iPSC-derived RPE cells with and without a p.Ser163Arg variant in CTRP5 that causes autosomal dominant late-onset retinal degeneration. After a thorough review, this reviewer does believe that this manuscript is appropriate for placement in Nature Communications Biology and will be a solid contribution to the eye field. That said, I do have a few comments that I would like the authors to digest and seriously consider before recommending publication:

Major Comments

1. In Figure 1D there is a 2-fold discrepancy in the N assessed for transepithelial resistance (N=243 for Healthy-iRPE versus N=121 for L-ORD-iRPE). Why is this? I think it is pretty obvious that if the N for L-ORD-iRPE equaled that of Healthy-iRPE that this would be a significant result. Why this discrepancy? Why did the authors not notice this? Please explain.
2. How many total images is Figure 2D representative of? To me this seems to be one of the most obvious and significant observations in the study. In my opinion, the authors have missed an obvious opportunity to perform CRISPR-Cas9-mediated correction of the p.Ser163Arg mutation in patient iPSCs. These CRISPR-corrected isogenic lines would then serve as even better control lines than their unaffected sibling controls due to the genetic matching between the lines and would serve as perfect controls for assessing the retainment of CTRP5 within early endosomes (Figure 2D) as well as constitutively-active downstream AMPK signaling. Moreover, because the p.Ser163Arg variant seems to be a harmful gain-of-function mutation, this presents an ideal situation for designing CRISPR reagents that are specific for the mutant allele only, especially since CTRP5 haploinsufficiency is not reported to cause retinal degeneration.
3. In the results section (page 7, paragraph 2, lines 228-230) the authors state that "Inhibition of AMPK activity in L-ORD-iRPE alleviated sub-RPE APOE deposits, a key hallmark of L-ORD and AMD" and reference this finding in Figure 3E. However, this finding was not significant and this insignificance was pointed out by the authors, which leads me to ask why the authors then focus on and state that reduction of AMPK signaling decreases a "key hallmark of L-ORD"? In other words, are the readers supposed to be impressed by the decrease of a non-significant finding?
4. Finally, the comparison of their findings from in vitro disease modelling of a dominant monogenic disease to that of a multigenic disease like AMD seems like a stretch to me, especially considering recent evidence in the last decade or so from multiple groups that demonstrate that AMD disease onset actually occurs in the choriocapillaris, not the RPE. Unless the authors are willing to generate patient iPSC-derived RPE from several AMD patients (hetero and homozygous for the CFH risk alleles) and compare them to that of L-ORD-iRPE side-by-side, this reviewer thinks that this portion of the manuscript should be removed, leaving the focus of the paper on CTRP5-associated L-ORD.

Minor Comments

1. Gene symbols should be italicized throughout the manuscript, especially in the title. "CTRP5 variant" should be "CTRP5 variant".
2. In Supplemental Figure 3E, it appears to my eye that ICC labeling for ADIPOR1 (green) is increased in L-ORD-iRPE compared to Healthy-iRPE. While I realize that this is sometimes hard to determine via fluorescent microscopy, I urge the authors to perform a Western blot for ADIPOR1 to see if this observation has any grounds for expansion in the manuscript.
3. In the results section (page 5, paragraph 2, line 157) the authors refer to "Figure S3F-G", but Supplemental Figure 3 only contains panels A-F. Please clarify this confusing figure reference.

Reviewer #2 (Remarks to the Author):

This manuscript investigates pathological mechanisms underlying mutant CTRP5 function in L-ORD. The authors use human iPSC-derived RPE cells to demonstrate that in L-ORD iRPE, constitutive activation of AMPK disrupts cellular metabolism/energy homeostasis, changes apical/basal VEGF secretion, and increases ApoE deposits. They show that Metformin treatment corrects the phenotypes associated with L-ORD iRPE. Because AMD and L-ORD share many phenotypes, understanding the molecular basis of some of the pathogenic features of L-ORD is of high importance. There is a wealth of significant data in this ms, and the experiments are rigorously performed for the most part. A few key issues that must be addressed are listed below:

1. Localization of mutant CTRP5 – early endosomes function in the endocytic/recycling pathway and not in the biosynthetic pathway. The presumed colocalization of mutant CTRP5 with EEA1 should be further discussed given that other studies show that it is retained in the ER (Shu et al., 2006). The images in Figures 2D and S3 showing colocalization with EEA1 and ER are of low quality. Better, higher magnification images are required to support the authors' claim of colocalization.
2. Adipor1: Have the authors performed IP of CTRP5 and adipor1 to show that these proteins do indeed interact? Adiponectin, the natural ligand of ADIPOR1, is present in both retina and RPE-Choroid - do both ligands bind to the receptor with equal affinity?
3. ApoE expression in L-ORD – westerns are required to show increased expression in L-ORD lines.
4. pAMPK is known to phosphorylate PGC1-alpha, which facilitates its deacetylation by SIRT1 leading to mitochondrial biogenesis. Despite the increased pAMPK levels, the authors have shown decreased mitochondrial function which they have attributed to decreased PLA2 activity. What is the status of PGC1-alpha in these cells?
5. Metformin treatment – this is very interesting, but brings up a few questions. First, metformin activates AMPK; and the authors hypothesize that mutant CTRP5 causes constitutive AMPK activation. So one would expect metformin to worsen the phenotype. Although few potential theories are presented in the discussion, the authors should explain how the duration and dose of metformin was chosen? Did the authors test other doses? Second, what is the effect of metformin treatment on mitochondrial function, DHA and NPD1 levels in L-ORD iRPE?
6. In Fig. 6 - in the metformin-treated cells, the decrease in basal ApoE appears to be offset by an increase in apical apoE? Better images would make this more convincing.
7. Statistics – it is unclear what the 'n' means in many of the figure legends – are these individual differentiations or data from different wells of the same differentiation? Were these technical replicates or true biological replicates?

Reviewer #3 (Remarks to the Author):

This manuscript is centered on the pathology of L-ORD from a unique perspective, using patient-derived iRPE cells as tools to understand the mechanisms causing the disease. The generation and characterization of these induced pluripotent stem cells-derived RPE from L-ORD patients is valuable and well presented in the manuscript. Data provided on their characterization is relevant for understanding the mechanisms of disease, and their generation is very useful for the field. Unfortunately, some aspects in the manuscript weaken the story, as there is not enough evidence provided to support mechanistic statements. The manuscript will benefit from revising the text accordingly, and/or providing scientific evidence to support certain aspects.

1. The hypothesis that CTRP5 interacts with ADIPOR1 is reasonable. However, it is very difficult to accept this hypothesis as fact based on colocalization evidence provided in the paper. The abstract

states that “we show that the dominant pathogenic CTRP5 variant causes disease by diminishing the apical secretion of CTRP5 and its binding affinity to adiponectin receptor 1”. This statement needs to be revised, to take into consideration that the manuscript does not fully demonstrate that binding between CTRP5 and ADIPOR1 occurs, or that the affinity of CTRP5 in this interaction is indeed decreased.

Also in the abstract it is mentioned that “These metabolic defects result in accumulation of sub-RPE drusen” There is still no proof that drusen in AMD is equivalent to deposits in L-ORD. Perhaps the authors need to rephrase this sentence by using the word sub-RPE deposits.

2. Lines 70-73 mention that “Mechanistically, reduced secretion of mutant/WT CTRP5 heterooligomers and their lower binding affinity to ADIPOR1 receptor is the likely reason for the genetically dominant behavior of this disease. Lower activity of mutant/WT CTRP5 heterooligomers result in constitutively higher AMPK activity leading to its insensitivity to changes in the cellular energy status.”

This paragraph is unclear, and needs to be revised. There is no direct proof that there is reduced secretion of hetero-oligomers provided by this study, only that there is reduced secretion in the total CTRP5 protein secretion from the apical side in iRPE cells from patients. Perhaps the S163R mutant remains trapped in the cell, and WT secretion alone is affected. Since WT and mutant are not tagged, no conclusion can firmly be drawn on the hetero-oligomers presence on the apical side. It is perhaps worth mentioning in the discussion that a previous in-vitro study in which WT and mutant S163R were coexpressed, both untagged, Stanton et al, 2017, Scientific Reports, noted that mutant decreases the total levels of secreted CTRP5.

It is also unclear what is meant by “lower activity of heterooligomers result in constitutively higher AMPK activity”. Specifically, what kind of activity do the oligomers display? How does the AMPK become constitutively active? The word “activity” is used twice in this sentence, leaving the readers confused about the meaning of this paragraph.

3. There is incomplete/incorrect information in connection with some references provided.

-For example, Lines 45 and 52 mention that “CTRP5 can assemble into heterologous higher-order multimeric complexes” and “Previously, mutant CTRP5 was shown to form heterooligomers with wildtype CTRP5”, ref 12, 19

The references provided do not directly demonstrate that CTRP5 forms hetero-oligomers. It is reasonable to assume that CTRP5 forms homo and heter-oligomers in patients, based on the behavior of other multimeric C1Q family members, and several papers on CTRP5 from previously published papers have made a similar reasonable assumption. However, references 12, 19 only show that CTRP5 is capable of self-assembling into multimeric structures. One specific reference directly supporting the existence of hetero-oligomers is the study by Shu, X. et al. Disease mechanisms in late-onset retinal macular degeneration associated with mutation in C1QTNF5. Hum Mol Genet (2006), which provided proof that hetero-oligomers form by using pull-down assays with epitope-tagged S163R and WT protein.

-Line 412, ref 66 appears to be incorrect. Ref 65 used in a previous sentence appears to be the correct one in this case, please check this and other references to back up the statements in the text “Furthermore, in RPE cells a strong co-labeling of CTRP5, specifically with ADIPOR1 and not ADIPOR2, was observed, suggesting a direct interaction between the two proteins - similar to CTRP9’s interaction with ADIPOR1 in the brain”66

-Lines 134-138: It is stated that “ELISA-based quantitative analysis showed L-ORD-iRPE CTRP5 apical and basal secretion was significantly lower compared to the healthy-iRPE (Ap: 14.3-fold, $p < 0.0001$;

135 Ba: 19.7-fold, $p < 0.0001$, L-ORD-iRPE vs healthy) in agreement with previously reported findings 32,33 (Figure 2B-C). This result also suggested that the mutant CTRP5 copy affects the secretion of the WT CTRP5 as well, likely through previously reported oligomerization between the WT and the mutant protein 12,19.”

Again, as mentioned above, references 12, 19 do not provide proof that WT and S163R form heterooligomers, they provide essential information on CTRP5 crystal structure, and on the fact that CTRP5 can multimerize into a bouquet-like octadecamer. The Ref. Shu et al 2006 needs to be additionally mentioned to support this statement.

-Within lines 395-400 it is stated that “Unlike recently reported overexpression of p.Ser163Arg CTRP5 in a mouse model, we do not observe the large, round intracellular globular aggregates of CTRP5. In fact, LORD-iRPE demonstrate relatively normal expression of RPE-specific genes and display typical monolayer transepithelial resistance. This is expected because the disease phenotypes or such deposits are not seen until the fourth or fifth decade of the patient’s life, suggesting that L-ORD-iRPE are a physiologically relevant model of the disease pathogenesis”

An essential aspect of the AAV-overexpression study (which the authors should mention when including this reference) is the formation of basal deposits consisting of CTRP5 protein, as a direct result of the basolateral misrouting of CTRP5 S163R mutant, an observation subsequently confirmed by Stanton et al in 2017 in their in-vitro study. Do the authors detect any basal accumulation of CTRP5 in their model? Although in their model the S163 and WT are not distinguishable when simultaneously expressed in LORD iRPE, it would be important to mention if there is a predominant apical or basal distribution in healthy versus patient iRPE CTRP5 levels.

4. Lines 139-141: It is stated that “The lower detection of secreted CTRP5 from L-ORD-iRPE was not due to Insolubility as shown in the Western blot of the protein pellet separated from the conditioned media by centrifugation (Figure S3A)” This figure is confusing. Panels A and B are unclear. In figure S3 legend it is stated: “Related to Figure 2. (A) Western blot of the pellet obtained from the apical media from healthy and L-ORD-iRPE after centrifugation demonstrate that the reduction in CTRP5 detected by ELISA is not due to mutant CTRP5 being insoluble and remaining in the pellet” Assuming the 30kDa bands are indeed CTRP5, the legend states the signal represents CTRP5 from the pellet fraction. It appears there is a lot in the pellet in the WT healthy and less in the patient. Did the authors measure the secreted CTRP5 in the media as well? Is this figure supposed to represent the secreted fraction in apical media, and not the pellet obtained from media centrifugation? In panel B there seems to be no CTRP5 present in lysates, which is unusual. The size of MW marker bands need to be included in the figure.

5. Lines 160-163: It is stated that: “Native immunogold labeling of healthy-iRPE further confirmed CTRP5 (12 nm gold particle) and ADIPOR1 (6 nm gold particle) interaction, as indicated by black arrows (Figure 2F). These co-labeling experiments suggest that apically secreted CTRP5 interacts with ADIPOR1 on the RPE surface and may modulate its activity.”

The figure is not clear, and the colocalization experiment is a weak proof that the two proteins interact. Other biochemical evidence, such as pull-down assays with tagged proteins, would provide a definitive proof that ADIPOR1 interacts with CTRP5. It is tempting to speculate this interaction is indeed real, based on a suggested colocalization (antibody validation against CTRP5 would be important here), and the fact that adiponectin is known to interact with this receptor. However, many other proteins are present at the apical RPE microvilli, and colocalization does not demonstrate an interaction, although it is an attractive and logical hypothesis.

6. Lines 424-427: "Here we show that in healthy-RPE, native CTRP5 acts as an inhibitor for ADIPOR1 receptors, since apical supplementation of gCTRP5 resulted in reduced pAMPK activity. In L-ORD, reduced availability of mutant/WT CTRP5 and reduced binding affinity of mutant/WT CTRP5 multimers leads to chronic activation of ADIPOR1. These results are further confirmed by the significantly higher constitutive AMPK phosphorylation seen in L-ORD"

This paragraph is not fully supported by evidence. Do the reduced levels in CTRP5 secretion increase the ADIPOR1 activity, and how is the receptor activity measured? What does the chronic activation of ADIPOR1 mean with respect to AMPK, is this receptor known to cause AMPK activation directly? Is ADIPOR1 also expressed in photoreceptor cells, and how could the hypothesized interaction with CTRP5 impact the disease in this case? Furthermore, is it possible that the addition of gCTRP5 modulates pAMPK levels through a distinct mechanism, unrelated to ADIPOR1 receptor?

7. On Line 228, it is mentioned "Inhibition of AMPK activity in L-ORD-iRPE alleviated sub-RPE APOE deposits, a key hallmark of L-ORD, and AMD (Figure 3E)." Fig 3E is not clear. Where are the apoE deposits localized? The basal side (green) seems to be localized opposite from the ApoE signal, separated by RPE nuclei. Better quality images will help to visualize the existence of sub-RPE deposits in iRPE cells from patients and define their localization. For how long was the AMPK activity inhibited to prevent the deposit formation? Is the ApoE accumulation reversible?

Reviewers' comments:

Reviewer #1 (Remarks to the Author):

Review of "AMPK modulation ameliorates dominant disease phenotypes of CTRP5 variant in retinal degeneration".

This is a very well-written and well-thought out and experimentally-executed manuscript demonstrating disease modelling of sibling patient iPSC-derived RPE cells with and without a p.Ser163Arg variant in CTRP5 that causes autosomal dominant late-onset retinal degeneration. After a thorough review, this reviewer does believe that this manuscript is appropriate for placement in Nature Communications Biology and will be a solid contribution to the eye field. That said, I do have a few comments that I would like the authors to digest and seriously consider before recommending publication:

Major Comments

1. In Figure 1D there is a 2-fold discrepancy in the N assessed for transepithelial resistance (N=243 for Healthy-iRPE versus N=121 for L-ORD-iRPE). Why is this? I think it is pretty obvious that if the N for L-ORD-iRPE equaled that of Healthy-iRPE that this would be a significant result. Why this discrepancy? Why did the authors not notice this? Please explain.

Re: The authors thank the reviewer for their rightful assessment that the TER measurements be re-evaluated. We have now added additional TER measurements to Figure 1G for both healthy-iRPE (N=266) and L-ORD-iRPE (N=264) to balance the sample sizes more appropriately. As the reviewer suggested, the TER of L-ORD-iRPE is significantly higher compared to Healthy-iRPE ($p < 0.0001$) and this has now been indicated in the text.

2. How many total images is Figure 2D representative of? To me this seems to be one of the most obvious and significant observations in the study. In my opinion, the authors have missed an obvious opportunity to perform CRISPR-Cas9-mediated correction of the p.Ser163Arg mutation in patient iPSCs. These CRISPR-corrected isogenic lines would then serve as even better control lines than their unaffected sibling controls due to the genetic matching between the lines and would serve as perfect controls for assessing the retainment of CTRP5 within early endosomes (Figure 2D) as well as constitutively-active downstream AMPK signaling. Moreover, because the p.Ser163Arg variant seems to be a harmful gain-of-function mutation, this presents an ideal situation for designing CRISPR reagents that are specific for the mutant allele only, especially since CTRP5 haploinsufficiency is not reported to cause retinal degeneration.

We thank the reviewer for highlighting the endosomal compartment retainment of mutant CTRP5. As reviewer suggested we performed additional experiments to better understand this phenomenon. We determined the Pearson's correlation coefficient between EEA1 and CTRP5 channels in higher magnification images and found that there was no significant difference between healthy and L-ORD cells (Fig. S3E; healthy: 0.22 ± 0.02 ; L-ORD 0.21 ± 0.3). We also noted areas where CTRP5 didn't colocalize with EEA1. Further analysis revealed that intracellular CTRP5 is colocalized in additional endo-lysosomal compartments. It colocalized with both ATG5 (Fig. 2D), a marker for mid-stage endo-lysosomes and LAMP1 (Fig. S3F), a marker for mature lysosomes. Interestingly, in both cases, higher CTRP5 accumulation was seen when cells were treated with Bafilomycin A1 (BafA1), a specific V-ATPase

inhibitor that disrupts lysosomal pH blocking both lysosomal activity and autophagic flux. Furthermore, we noticed a difference in Pearson's correlation coefficient, for ATG5 and CTRP5 colocalization, between healthy and L-ORD cells untreated ($p < 0.05$) and treated with Bafilomycin A1 ($p < 0.01$) (healthy without BafA1: 0.07 ± 0.02 ; with BafA1: 0.16 ± 0.02 ; L-ORD without BafA1: 0.16 ± 0.04 ; with BafA1 0.37 ± 0.08). Similarly, Pearson's correlation coefficient for Bafilomycin A1 untreated and treated healthy and L-ORD cells for CTRP5 and LAMP1 showed a 35% higher correlation in the presence of Bafilomycin A1, suggesting accumulation of endo-lysosomal targeted proteins. Overall, these results suggested that mutant CTRP5 is targeted for degradation via the endo-lysosomal compartments inside the cell. This observation is consistent with the known mechanism of protein aggregate degradation in neurodegenerative diseases (Monaco and Fraldi 2020).

We performed an additional experiment to support the idea that mutant CTRP5 traps WT CTRP5 inside cells. We co-expressed V5-tagged WT and FLAG-tagged mutant CTRP5 in healthy RPE and noticed a dose-dependent entrapment of WT-CTRP5 as specifically mutant CTRP5 levels were increased. When low levels (MOI 0.5) of V5-tagged WT CTRP5 expressing lentivirus were co-transduced with low levels (MOI 0.5) of FLAG-tagged mutant CTRP5, we detected low intracellular signal for the WT protein. But while keeping the levels of V5-tagged WT CTRP5 expressing lentivirus the same (MOI 0.5), when we increased levels of FLAG-tagged mutant CTRP5 expressing lentivirus (MOI 3.0), we noticed significantly higher levels of WT-CTRP5 trapped inside the cell (Figure 2D). Same entrapment of WT-CTRP5 was not seen when only the levels of WT-CTRP expressing lentivirus were increased to the same extent (Figure S3B). We hypothesize that as more WT CTRP5 make heterooligomers with mutant CTRP5, more entrapment is seen inside the cell.

We agree with the reviewer's suggestion of performing CRISPR/Cas9 mediated correction of the S163R mutation to ameliorate L-ORD phenotype cells. We tried making this correction, however, this locus was not targetable - in our hands. We then resorted to a slightly different gene therapy approach. Recently, it was demonstrated dominant ocular disease phenotype can be corrected in iRPE cells using a gene augmentation approach (Sinha et al. 2020). We tested this idea in L-ORD-iRPE by overexpressing WT-CTRP5 using lentivirus. This data is now presented in Figure S6. Here we show that overexpression of WT-CTRP5 in L-ORD-iRPE corrects the apical and basal CTRP5 secretion, inhibits AMPK activation, and rescues mis-polarization of VEGF. Overall, this data not only provides further confidence that cellular disease phenotype seen in our model are triggered by reduced CTRP5 levels, it also provides proof-of-concept gene therapy approach for L-ORD patients. Lentivirus based gene therapy vectors are in clinical use in immuno-oncology (Milone and O'Doherty 2018)).

We have also provided discussion on this topic.

3. In the results section (page 7, paragraph 2, lines 228-230) the authors state that "Inhibition of AMPK activity in L-ORD-iRPE alleviated sub-RPE APOE deposits, a key hallmark of L-ORD and AMD" and reference this finding in Figure 3E. However, this finding was not significant and this insignificance was pointed out by the authors, which leads me to ask why the authors then focus on and state that reduction of AMPK signaling decreases a "key hallmark of L-ORD"? In other words, are the readers supposed to be impressed by the decrease of a non-significant finding?

Re: To understand the effect of AMPK inhibition on subRPE APOE deposits, we have performed additional testing via cryo-sectioning of transwells to better view apical vs subRPE deposits. Our new data clearly

demonstrates that AMPK inhibition (ara-A) reduces APOE expression and has a prominent effect on subRPE APOE accumulation. New data is now provided in Figure 3I. We apologize for not stating this clearly previously. It is now noted in the text that AMPK inhibition indeed reduces subRPE APOE deposits.

4. Finally, the comparison of their findings from in vitro disease modelling of a dominant monogenic disease to that of a multigenic disease like AMD seems like a stretch to me, especially considering recent evidence in the last decade or so from multiple groups that demonstrate that AMD disease onset actually occurs in the choriocapillaris, not the RPE. Unless the authors are willing to generate patient iPSC-derived RPE from several AMD patients (hetero and homozygous for the CFH risk alleles) and compare them to that of L-ORD-iRPE side-by-side, this reviewer thinks that this portion of the manuscript should be removed, leaving the focus of the paper on CTRP5-associated L-ORD.

Re: We thank the reviewer for expressing their concern regarding the relevance of extrapolating the findings from this study (L-ORD) to AMD. We have removed most extrapolation to AMD in the manuscript including the retrospective clinical data describing the use of metformin in AMD patients.

Minor Comments

1. Gene symbols should be italicized throughout the manuscript, especially in the title. “CTRP5 variant” should be “*CTRP5* variant”.

Re: We have made this correction to the title and throughout the manuscript.

2. In Supplemental Figure 3E, it appears to my eye that ICC labeling for ADIPOR1 (green) is increased in L-ORD-iRPE compared to Healthy-iRPE. While I realize that this is sometimes hard to determine via fluorescent microscopy, I urge the authors to perform a Western blot for ADIPOR1 to see if this observation has any grounds for expansion in the manuscript.

Re: We thank the reviewer for this observation and have explored this further. We now provide additional data in Figures S3H-J. Our data shows that ADIPOR1 expression is higher in healthy iRPE as compared to L-ORD-iRPE. Figure S3H, qPCR analysis shows gene expression for ADIPOR1 revealing a 1.9-fold higher expression in healthy-iRPE compared to L-ORD ($p < 0.05$). Consistent with the decrease in transcript we now provide a Western blot (Figure S3I) and its quantification (Figure S3J) showing a 1.8-fold higher expression of ADIPOR1 in healthy-iRPE compared to L-ORD ($p < 0.05$). The decrease in ADIPOR1 expression in L-ORD-iRPE could be interpreted as a cellular response to compensate for constitutively active AMPK.

3. In the results section (page 5, paragraph 2, line 157) the authors refer to “Figure S3F-G”, but Supplemental Figure 3 only contains panels A-F. Please clarify this confusing figure reference.

Re: This reference has been corrected

Reviewer #2 (Remarks to the Author):

This manuscript investigates pathological mechanisms underlying mutant CTRP5 function in L-ORD. The authors use human iPSC-derived RPE cells to demonstrate that in L-ORD iRPE, constitutive activation of AMPK disrupts cellular metabolism/energy homeostasis, changes apical/basal VEGF secretion, and increases ApoE deposits. They show that Metformin treatment corrects the phenotypes associated with L-ORD iRPE. Because AMD and L-ORD share many phenotypes, understanding the molecular basis of some of the pathogenic features of L-ORD is of high importance. There is a wealth of significant data in this ms, and the experiments are rigorously performed for the most part. A few key issues that must be addressed are listed below:

1. Localization of mutant CTRP5 – early endosomes function in the endocytic/recycling pathway and not in the biosynthetic pathway. The presumed colocalization of mutant CTRP5 with EEA1 should be further discussed given that other studies show that it is retained in the ER (Shu et al., 2006). The images in Figures 2D and S3 showing colocalization with EEA1 and ER are of low quality. Better, higher magnification images are required to support the authors' claim of colocalization.

We thank the reviewer for highlighting the endosomal compartment retainment of mutant CTRP5. As reviewer suggested we performed additional experiments to better understand this phenomenon. We determined the Pearson's correlation coefficient between EEA1 and CTRP5 channels in higher magnification images and found that there was no significant difference between healthy and LORD cells (Fig. S3E; healthy: 0.22 ± 0.02 ; L-ORD 0.21 ± 0.3). We also noted areas where CTRP5 didn't colocalize with EEA1. Further analysis revealed that intracellular CTRP5 is colocalized in additional endo-lysosomal compartments. It colocalized with both ATG5 (Fig. 2D), a marker for mid-stage endo-lysosomes and LAMP1 (Fig. S3F), a marker for mature lysosomes. Interestingly, in both cases, higher CTRP5 accumulation was seen when cells were treated with Bafilomycin A1 (BafA1), a specific V-ATPase inhibitor that disrupt lysosomal pH blocking both lysosomal activity and autophagic flux. Furthermore, we noticed a difference in Pearson's Correlation Coefficient, for ATG5 and CTRP5 colocalization, between healthy and L-ORD cells untreated ($p < 0.05$) and treated with Bafilomycin A1 ($p < 0.01$) (healthy without BafA1: 0.07 ± 0.02 ; with BafA1: 0.16 ± 0.02 ; L-ORD without BafA1: 0.16 ± 0.04 ; with BafA1 0.37 ± 0.08). Similarly, Pearson's Correlation Coefficient for Bafilomycin A1 untreated and treated healthy and L-ORD cells for CTRP5 and LAMP1 showed a 35% higher correlation in the presence of Bafilomycin A1, suggest accumulation of endo-lysosomal targeted proteins. Overall, these results suggested that mutant CTRP5 is targeted for degradation via the endo-lysosomal compartments inside the cell. This observation is consistent with the known mechanism of protein aggregate degradation in neurodegenerative diseases (Monaco and Fraldi 2020).

We performed an additional experiment to support the idea that mutant CTRP5 traps WT CTRP5 inside cells. We co-expressed V5-tagged WT and FLAG-tagged mutant CTRP5 in healthy RPE and noticed a dose-dependent entrapment of WT-CTRP5 as specifically mutant CTRP5 levels were increased. When low levels (MOI 0.5) of V5-tagged WT CTRP5 expressing lentivirus were cotransduced with low levels (MOI 0.5) of FLAG-tagged mutant CTRP5, we detected low intracellular signal for the WT protein. But while keeping the levels of V5-tagged WT CTRP5 expressing lentivirus the same (MOI 0.5), when we increased levels of FLAG-tagged mutant CTRP5 expressing lentivirus (MOI 3.0), we noticed significantly higher levels of WT-CTRP5 trapped inside the cell (Figure 2D). Same entrapment of WT-CTRP5 was not seen when only the levels of WT-CTRP expressing lentivirus were increased to the same extent (Figure S3B).

We hypothesize that as more WT CTRP5 form heterooligomers with mutant CTRP5, more entrapment is seen inside the cell.

We have also provided discussion on this topic.

2. Adipor1: Have the authors performed IP of CTRP5 and adipor1 to show that these proteins do indeed interact? Adiponectin, the natural ligand of ADIPOR1, is present in both retina and RPE-Choroid - do both ligands bind to the receptor with equal affinity?

Re: We thank the reviewer for pointing this out and we now provide Co-IP to show that CTRP5 and ADIPOR1 do indeed interact (Figure 2G). With regards to adiponectin, the natural ligand of ADIPOR1, we refer the reviewer to Figure 3B. Our RPE cell culture media contains 5% FBS, which has been shown to contain high levels of adiponectin (Wang et al. 2004). In Figure 3B and C, we measured the effects of recombinant CTRP5 (0.2 µg/ml, globular head) on pAMPK levels and found that the effect of CTRP5 was partially masked by the presence of serum. This masking effect was likely due to adiponectin in serum having a higher binding affinity to ADIPOR1 or that it was present at much higher concentrations than CTRP5 under these culture conditions. However, under serum free conditions, adding recombinant CTRP5 caused a 20% reduction in pAMPK levels in healthy-iRPE. This inhibition of pAMPK levels was notably absent in L-ORD-iRPE because, as we now show in Figure 2C, mutant CTRP5 traps WT CTRP5 within the cell.

3. ApoE expression in L-ORD – westerns are required to show increased expression in L-ORD lines.

Re: We thank the reviewer for this suggestion and we now provide WB of APOE in healthy and L-ORD-iRPE cell lysates before and after metformin treatment (Figure 6D; quantification in 6E). This result shows that the total amount of cellular APOE remains unchanged. However, IF images show a difference in APOE localization (Figure 6B and Figure 6C lower panel) with subcellular APOE being trapped in the transwell membrane (not accounted for in the WB of cell lysates).

In L-ORD-iRPE, APOE accumulates below the RPE (APOE colocalizes with Collagen IV, an RPE basal marker and is trapped underneath and within the transwell. Metformin treatment reduces the APOE signal co-localizing with Collagen IV and alleviates APOE accumulation underneath and within the transwell.

4. pAMPK is known to phosphorylate PGC1-alpha, which facilitates its deacetylation by SIRT1 leading to mitochondrial biogenesis. Despite the increased pAMPK levels, the authors have shown decreased mitochondrial function which they have attributed to decreased PLA2 activity. What is the status of PGC1-alpha in these cells?

Re: We thank the reviewer for this insight and we now provide western blots and quantification of both PGC1α (Figure 3E-F) and Phospho (Ser-570)-PGC1α (Figure 3G-H). The representative Western blot of PGC1α shows no difference in expression in healthy (n=4) and L-ORD-iRPE (n=4). However, the representative Western blot of Phospho (Ser-570)-PGC1α shows a significant increase (p=0.03) in L-ORD-iRPE (n=4) compared to healthy-iRPE (n=4). Ser 570 phosphorylation is linked to inhibition of PGC1α transcriptional activity (Fernandez-Marcos and Auwerx 2011). Based on these results, we conclude that

decreased mitochondrial function is also due to reduced transcriptional activity of PGC1 α reducing mitochondrial biogenesis.

5. Metformin treatment – this is very interesting, but brings up a few questions. First, metformin activates AMPK; and the authors hypothesize that mutant CTRP5 causes constitutive AMPK activation. So one would expect metformin to worsen the phenotype. Although few potential theories are presented in the discussion, the authors should explain how the duration and dose of metformin was chosen? Did the authors test other doses? Second, what is the effect of metformin treatment on mitochondrial function, DHA and NPD1 levels in L-ORD iRPE?

Re: We thank the reviewer for their interest in metformin and other AMPK modulators as a potential treatment.

AMPK is considered a double-edged sword. While an acute activation of AMPK is linked to improved cellular metabolism, chronic AMPK activation has been linked to diseases such as diabetes (Yavari et al. 2016) and neurodegeneration in Huntington (Ju et al. 2011). We believe a similar phenomenon happens in L-ORD-iRPE. There is a sustained (from birth) activation of AMPK in L-ORD-iRPE cells, as shown in Figure 3A. This leads to inability of L-ORD-iRPE cells to respond to any metabolic change, as shown in Figure 3D. Hence a drop in mitochondrial activities including mitochondrial respiration and fatty acid oxidation (Figures 4F-H). As a compensatory response, glycolysis increases inducing RPE dedifferentiation and atrophy over time (Figures 5A, B).

We believe metformin acts in L-ORD-iRPE indirectly on AMPK. It has been shown that metformin is a weak but a specific inhibitor of mitochondrial complex I (Vial, Demaille and Guigas 2019). This results in mild toxicity and as a compensatory mechanism to this toxicity, cells undergo mitochondrial biogenesis (Loubiere et al. 2017) which switches cellular metabolism back from glycolytic to oxidative phosphorylation, correcting both fatty acid metabolism and dedifferentiation. Consistently, metformin is shown to suppress epithelial dedifferentiation via suppressing glycolysis and activation of mitochondrial respiration (Del Barco et al. 2011).

Both of these theories with appropriate references have been added to the discussion.

AMPK is a central player downstream of the dominant pathogenic effect of mutant CTRP5 variant. To prove this, we now more clearly show that ara-A, an AMPK activity inhibitor, ameliorates subRPE deposits seen in L-ORD-iRPE.

The 3mM metformin dose was selected because it is a commonly used concentration for in vitro studies (Zhao et al. 2020, Qu et al. 2020).

With respect to the duration of metformin use, we found that with 2 weeks metformin (3mM) treatment the apical/basal VEGF polarity could be completely restored. Any time lower than 2 weeks didn't have significant effect on VEGF polarity defect.

We now provide two additional data demonstrating that metformin is able to improve fatty acid oxidation function of mitochondria. There is about a 25% increase in levels of β -hydroxybutyrate, which is formed via fatty acid oxidation of acetyl-CoA in metformin treated L-ORD-iRPE (Figure 5G). Similarly,

DHA derived NPD1 levels are increased by about 40% in metformin treated L-ORD-iRPE (Figure 5H). Both these data confirm that metformin is able to improve mitochondrial activity in L-ORD-iRPE and also supports our hypothesis that mitochondrial fatty acid oxidation, not AMPK, is the primary target of metformin.

6. In Fig. 6 - in the metformin-treated cells, the decrease in basal ApoE appears to be offset by an increase in apical apoE? Better images would make this more convincing.

Re: In response to the reviewer's comment for improved image quality we now provide an update to Figures 6A-C. Reduced subRPE APOE deposits are apparent in metformin treated samples. Higher apical APOE are not apparent but that's likely due to the fact that immunofluorescent images are not quantitative. We provide an additional experiment of L-ORD-iRPE treated with ara-A (AMPK inhibitor) presented in Figure 3I and see similar outcome of reduced subRPE APOE upon AMPK inhibition.

7. Statistics – it is unclear what the 'n' means in many of the figure legends – are these individual differentiations or data from different wells of the same differentiation? Were these technical replicates or true biological replicates?

Re: The n is specified in the main text in the first paragraph of Results: "Thus, each experiment presented here uses averaged data from iRPE derived from four iPSC clones of two unaffected siblings (healthy-iRPE) and four iPSC clones of two patients (L-ORD-iRPE)."

Reviewer #3 (Remarks to the Author):

This manuscript is centered on the pathology of L-ORD from a unique perspective, using patient-derived iRPE cells as tools to understand the mechanisms causing the disease. The generation and characterization of these induced pluripotent stem cells-derived RPE from L-ORD patients is valuable and well presented in the manuscript. Data provided on their characterization is relevant for understanding the mechanisms of disease, and their generation is very useful for the field. Unfortunately, some aspects in the manuscript weaken the story, as there is not enough evidence provided to support mechanistic statements. The manuscript will benefit from revising the text accordingly, and/or providing scientific evidence to support certain aspects.

1. The hypothesis that CTRP5 interacts with ADIPOR1 is reasonable. However, it is very difficult to accept this hypothesis as fact based on colocalization evidence provided in the paper. The abstract states that "we show that the dominant pathogenic CTRP5 variant causes disease by diminishing the apical secretion of CTRP5 and its binding affinity to adiponectin receptor 1". This statement needs to be revised, to take into consideration that the manuscript does not fully demonstrate that binding between CTRP5 and ADIPOR1 occurs, or that the affinity of CTRP5 in this interaction is indeed decreased.

Re: We acknowledge the reviewer's concern for more evidence of CTRP5 and ADIPOR1 interaction. To this end we now include Co-IP showing direct interaction between these two proteins (Figure 2G). This

data together with the previous immuno-SEM strongly supports our hypothesis of direct interactions between ADIPOR1 and CTRP5.

We performed an additional experiment to support the idea that mutant CTRP5 traps WT CTRP5 inside cells. We co-expressed V5-tagged WT and FLAG-tagged mutant CTRP5 in healthy RPE and noticed a dose-dependent entrapment of WT-CTRP5 as specifically mutant CTRP5 levels were increased. When low levels (MOI 0.5) of V5-tagged WT CTRP5 expressing lentivirus were co-transduced with low levels (MOI 0.5) of FLAG-tagged mutant CTRP5, we detected low intracellular signal for the WT protein. But while keeping the levels of V5-tagged WT CTRP5 expressing lentivirus the same (MOI 0.5), when we increased levels of FLAG-tagged mutant CTRP5 expressing lentivirus (MOI 3.0), we noticed significantly higher levels of WT-CTRP5 trapped inside the cell (Figure 2D). Same entrapment of WT-CTRP5 was not seen when only the levels of WT-CTRP5 expressing lentivirus were increased to the same extent (Figure S3B). We hypothesize that as more WT CTRP5 form heterooligomers with mutant CTRP5, more entrapment is seen inside the cell.

We have also amended the abstract to now read, “we show that the dominant pathogenic CTRP5 variant leads to reduced apical CTRP5 secretion. In silico modeling suggests lower binding of mutant CTRP5 to adiponectin receptor 1 (ADIPOR1).”

Also in the abstract it is mentioned that “These metabolic defects result in accumulation of sub-RPE drusen” There is still no proof that drusen in AMD is equivalent to deposits in L-ORD. Perhaps the authors need to rephrase this sentence by using the word sub-RPE deposits.

Re: We thank the reviewer for making this distinction. We have removed AMD-like comparisons from the main text and now focus solely on the disease mechanism as it relates to L-ORD. We have also amended the abstract as per the reviewer’s request, “sub-RPE deposits” in place of “drusen”.

2. Lines 70-73 mention that “Mechanistically, reduced secretion of mutant/WT CTRP5 heterooligomers and their lower binding affinity to ADIPOR1 receptor is the likely reason for the genetically dominant behavior of this disease. Lower activity of mutant/WT CTRP5 heterooligomers result in constitutively higher AMPK activity leading to its insensitivity to changes in the cellular energy status.”

This paragraph is unclear, and needs to be revised. There is no direct proof that there is reduced secretion of hetero-oligomers provided by this study, only that there is reduced secretion in the total CTRP5 protein secretion from the apical side in iRPE cells from patients. Perhaps the S163R mutant remains trapped in the cell, and WT secretion alone is affected. Since WT and mutant are not tagged, no conclusion can firmly be drawn on the hetero-oligomers presence on the apical side. It is perhaps worth mentioning in the discussion that a previous in-vitro study in which WT and mutant S163R were coexpressed, both untagged, Stanton et al, 2017, Scientific Reports, noted that mutant decreases the total levels of secreted CTRP5.

Re: We agree with the reviewer that this paragraph requires clarity. Based on reviewer’s suggestion we performed an overexpression study in iRPE using WT (V5 tag) and Mutant CTRP5 (flag) proteins. These results show that mutant CTRP5 traps the WT protein inside the cell in a dose dependent manner. These

results are shown in Figure 2D. See response to question #1 (reviewer 3) for a detailed description of this experiment.

Also, based on reviewer's suggestion, the above paragraph is now changed to read, "Mechanistically, reduced apical secretion of WT-mutant CTRP5 heterooligomers and their predicted lower binding affinity to ADIPOR1 receptor is the likely reason for the genetically dominant behavior of this disease. We show that lower CTRP5 levels are associated with constitutively activated AMPK leading to its insensitivity to changes in the cellular energy status." We have also appropriately referenced Stanton et al 2017 for comparison and discussed new results in discussion.

It is also unclear what is meant by "lower activity of heterooligomers result in constitutively higher AMPK activity". Specifically, what kind of activity do the oligomers display? How does the AMPK become constitutively active? The word "activity" is used twice in this sentence, leaving the readers confused about the meaning of this paragraph.

We apologize for the confusing language. The activity here refers to the ability of CTRP5 to bind ADIPOR1 and suppress its activity, which downstream of ADIPOR1 is translated as activation of AMPK. Lower secretion of CTRP5 (Figures 2A, B), likely due the entrapment of WT CTRP5 by the mutant CTRP5 inside the cell (Figures 2C, D), and reduced predicted binding of mutant CTRP5 variant to ADIPOR1 receptor – all these combined together reflect lower ability (activity) of WT-mutant CTRP5 heterooligomers to suppress ADIPOR1 mediated AMPK activation – resulting in constitutive activation of AMPK in L-ORD-iRPE (Figure3A). We have clarified and correct the text.

3. There is incomplete/incorrect information in connection with some references provided.

-For example, Lines 45 and 52 mention that "CTRP5 can assemble into heterologous higher-order multimeric complexes" and "Previously, mutant CTRP5 was shown to form heterooligomers with wildtype CTRP5", ref 12, 19

The references provided do not directly demonstrate that CTRP5 forms hetero-oligomers. It is reasonable to assume that CTRP5 forms homo and heter-oligomers in patients, based on the behavior of other multimeric C1Q family members, and several papers on CTRP5 from previously published papers have made a similar reasonable assumption. However, references 12, 19 only show that CTRP5 is capable of self-assembling into multimeric structures. One specific reference directly supporting the existence of hetero-oligomers is the study by Shu, X. et al. Disease mechanisms in late-onset retinal macular degeneration associated with mutation in C1QTNF5. Hum Mol Genet (2006), which provided proof that hetero-oligomers form by using pull-down assays with epitope-tagged S163R and WT protein.

Re: We thank the reviewer for pointing this out. We have now included the reference suggested of Shu, X. et al (2006). To provide additional evidence that mutant CTRP5 may form heterooligomers with WT-CTRP5 we performed an overexpression study in iRPE using WT (V5 tag) and Mutant CTRP5 (flag) proteins. These results show that mutant CTRP5 traps the WT protein inside the cell in a dose dependent manner. These results are shown in Figure 2D. See response to question #1 (reviewer 3) for a detailed description of this experiment.

-Line 412, ref 66 appears to be incorrect. Ref 65 used in a previous sentence appears to be the correct one in this case, please check this and other references to back up the statements in the text “Furthermore, in RPE cells a strong co-labeling of CTRP5, specifically with ADIPOR1 and not ADIPOR2, was observed, suggesting a direct interaction between the two proteins - similar to CTRP9’s interaction with ADIPOR1 in the brain”⁶⁶

Re: We thank the reviewer and have corrected this reference. The correct reference as the reviewer pointed out is from Kambara (2015).

-Lines 134-138: It is stated that “ELISA-based quantitative analysis showed L-ORD-iRPE CTRP5 apical and basal secretion was significantly lower compared to the healthy-iRPE (Ap: 14.3-fold, $p < 0.0001$; 135 Ba: 19.7-fold, $p < 0.0001$, L-ORD-iRPE vs healthy) in agreement with previously reported findings 32,33 (Figure 2B-C). This result also suggested that the mutant CTRP5 copy affects the secretion of the WT CTRP5 as well, likely through previously reported oligomerization between the WT and the mutant protein 12,19.”

Again, as mentioned above, references 12, 19 do not provide proof that WT and S163R form heterooligomers, they provide essential information on CTRP5 crystal structure, and on the fact that CTRP5 can multimerize into a bouquet-like octadecamer. The Ref. Shu et al 2006 needs to be additionally mentioned to support this statement.

Re: We thank the reviewer for pointing this out. We have now included the suggested reference of Shu, X. et al (2006). Please also refer to the new experiment shown in Figure 3B that suggests formation of heterooligomers between WT and mutant CTRP5.

-Within lines 395-400 it is stated that “Unlike recently reported overexpression of p.Ser163Arg CTRP5 in a mouse model, we do not observe the large, round intracellular globular aggregates of CTRP5. In fact, LORD-iRPE demonstrate relatively normal expression of RPE-specific genes and display typical monolayer transepithelial resistance. This is expected because the disease phenotypes or such deposits are not seen until the fourth or fifth decade of the patient’s life, suggesting that L-ORD-iRPE are a physiologically relevant model of the disease pathogenesis”

An essential aspect of the AAV-overexpression study (which the authors should mention when including this reference) is the formation of basal deposits consisting of CTRP5 protein, as a direct result of the basolateral misrouting of CTRP5 S163R mutant, an observation subsequently confirmed by Stanton et al in 2017 in their in-vitro study. Do the authors detect any basal accumulation of CTRP5 in their model? Although in their model the S163 and WT are not distinguishable when simultaneously expressed in LORD iRPE, it would be important to mention if there is a predominant apical or basal distribution in healthy versus patient iRPE CTRP5 levels.

Re: We appreciate the reviewer’s comment on the localization of CTRP5. We have changed the paragraph (5th paragraph of Discussion) to directly address this: “Our data suggest that reduced secretion of CTRP5 is likely because oligomers of wildtype and mutant protein are trapped inside the cell. An increase in expression of the FLAG-tagged mutant CTRP5 increased the intracellular signal for V5-tagged WT CTRP5 in a dose-dependent manner suggesting the mutant variant traps the WT inside the cell and lowers its expression. The heterooligomers of WT-mutant CTRP5 are likely trapped in an endo-lysosomal compartment inside the cell. Pearson’s correlation coefficient clearly showed strongest co-labeling of CTRP5 with LAMP1/2 and ATG5, especially when lysosomal degradation is blocked using

bafilomycin. Our results differ from Shu et al that showed endoplasmic reticulum localization of WT and mutant CTRP5. This discrepancy is likely because Shu et al used an overexpression system of either WT or mutant CTRP5. A similar discrepancy exists between our data and another study looking at overexpression of p.Ser163Arg CTRP5 in a mouse model (Dinculescu et al. 2015). Dinculescu observed large, round intracellular globular aggregates of CTRP5 in this model (Dinculescu et al. 2015), which are not seen in our L-ORD-iRPE (Figure S3G). We are detecting endogenous expression of heterooligomers of WT-CTRP5 as opposed to overexpression of WT or mutant proteins that was done in an RPE cell line or an adult mouse RPE.”

4. Lines 139-141: It is stated that “The lower detection of secreted CTRP5 from L-ORD-iRPE was not due to Insolubility as shown in the Western blot of the protein pellet separated from the conditioned media by centrifugation (Figure S3A)” This figure is confusing. Panels A and B are unclear. In figure S3 legend it is stated: “Related to Figure 2. (A) Western blot of the pellet obtained from the apical media from healthy and L-ORD-iRPE after centrifugation demonstrate that the reduction in CTRP5 detected by ELISA is not due to mutant CTRP5 being insoluble and remaining in the pellet” Assuming the 30kDa bands are indeed CTRP5, the legend states the signal represents CTRP5 from the pellet fraction. It appears there is a lot in the pellet in the WT healthy and less in the patient. Did the authors measure the secreted CTRP5 in the media as well? Is this figure supposed to represent the secreted fraction in apical media, and not the pellet obtained from media centrifugation? In panel B there seems to be no CTRP5 present in lysates, which is unusual. The size of MW marker bands need to be included in the figure.

We apologize for confusing language. We have now removed this data and provide CTRP5 ELISA in Figures 2A and B. We clearly see reduced secretion of CTRP5 in L-ORD-iRPE as compared to healthy-iRPE.

5. Lines 160-163: It is stated that: “Native immunogold labeling of healthy-iRPE further confirmed CTRP5 (12 nm gold particle) and ADIPOR1 (6 nm gold particle) interaction, as indicated by black arrows (Figure 2F). These co-labeling experiments suggest that apically secreted CTRP5 interacts with ADIPOR1 on the RPE surface and may modulate its activity.”

The figure is not clear, and the colocalization experiment is a weak proof that the two proteins interact. Other biochemical evidence, such as pull-down assays with tagged proteins, would provide a definitive proof that ADIPOR1 interacts with CTRP5. It is tempting to speculate this interaction is indeed real, based on a suggested colocalization (antibody validation against CTRP5 would be important here), and the fact that adiponectin is known to interact with this receptor. However, many other proteins are present at the apical RPE microvilli, and colocalization does not demonstrate an interaction, although it is an attractive and logical hypothesis.

Re: We agree with the reviewer’s assessment and now provide Co-IP to show the interaction between ADIPOR1 and CTRP5 (Figure 2G). Together, immuno-fluorescent colocalization, immune-SEM, and Co-IP provide stronger evidence for a physical interaction between ADIPOR1 and CTRP5.

6. Lines 424-427: “Here we show that in healthy-RPE, native CTRP5 acts as an inhibitor for ADIPOR1 receptors, since apical supplementation of gCTRP5 resulted in reduced pAMPK activity. In L-ORD, reduced availability of mutant/WT CTRP5 and reduced binding affinity of mutant/WT CTRP5 multimers

leads to chronic activation of ADIPOR1. These results are further confirmed by the significantly higher constitutive AMPK phosphorylation seen in L-ORD”

This paragraph is not fully supported by evidence. Do the reduced levels in CTRP5 secretion increase the ADIPOR1 activity, and how is the receptor activity measured? What does the chronic activation of ADIPOR1 mean with respect to AMPK, is this receptor known to cause AMPK activation directly? Is ADIPOR1 also expressed in photoreceptor cells, and how could the hypothesized interaction with CTRP5 impact the disease in this case? Furthermore, is it possible that the addition of gCTRP5 modulates pAMPK levels through a distinct mechanism, unrelated to ADIPOR1 receptor?

We thank the reviewer for posing interesting questions and provide the answers below:

Do the reduced levels in CTRP5 secretion increase the ADIPOR1 activity, and how is the receptor activity measured?

ADIPOR1 is a ceramidase. We hypothesize that the reduction of CTRP5 levels will not increase the activity of ADIPOR1 because, in the absence of CTRP5, ADIPOR1 just remains constitutively active by ADIPONECTIN. CTRP5 is a negative regulator that fine-tunes the activity of ADIPOR1. Yamaguchi et al (DOI: 10.1038/nm1557) show that ADIPOR1 is constitutively activated by ADIPONECTIN and regulate the activity of AMPK inside cells. This reference is provided.

We have directly measured the ceramidase activity of this receptor now and find it unchanged as compared to healthy-iRPE (Figure S5D), consistently levels of ceramide were not different under the two conditions (Figures S5C).

In serum free media (devoid of CTRP5 and adiponectin) addition of CTRP5 decreases AMPK activity in healthy-iRPE (see Figure 3C), suggesting that CTRP5 regulates AMPK activity via ADIPOR1.

What does the chronic activation of ADIPOR1 mean with respect to AMPK, is this receptor known to cause AMPK activation directly?

Re: Yes, there is ample literature evidence supporting ADIPOR1 activates AMPK directly (Cammisotto and Bendayan 2008, Zheng et al. 2011). Some of these references are cited in the manuscript.

Is ADIPOR1 also expressed in photoreceptor cells, and how could the hypothesized interaction with CTRP5 impact the disease in this case?

Re: Yes, Dr. Bazan’s lab (a collaborator on this paper as well) has shown that photoreceptors do indeed express ADIPOR1 (Rice et al. 2015). In this work, Dr. Bazan’s group showed that ADIPOR1 also controls uptake of DHA. Ablation of AdipoR1 expression results in DHA reduction. In mice lacking ADIPOR1 visual function is reduced evidenced by attenuated electroretinogram responses and progressive photoreceptor degeneration. Our study reveals that CTRP5 helps regulate ADIPOR1’s AMPK activity which in turn can alter lipid metabolism and the release of free fatty acids needed to produce DHA and DHA derivatives such as neuroprotection D1 (NPD1). These factors are critical for maintaining photoreceptor survival.

Since photoreceptors do not express CTRP5, this study is yet another example how RPE cells regulate homeostasis across the RPE-photoreceptor interface.

We have now included Rice et al 2015 in the discussion.

Furthermore, is it possible that the addition of gCTRP5 modulates pAMPK levels through a distinct mechanism, unrelated to ADIPOR1 receptor?

Re: It remains possible that globular CTRP5 modulates pAMPK levels through a distinct mechanism in addition to or unrelated to ADIPOR1, although the Co-IP data strengthens our claims of direct interactions between CTRP5 and ADIPOR1 (Figure 2G) and suggest that, if at all, the action of gCTRP5 regulating pAMPK is a mechanism additional (not an unrelated) to CTRP5 regulating ADIPOR1 activity.

7. On Line 228, it is mentioned "Inhibition of AMPK activity in L-ORD-iRPE alleviated sub-RPE APOE deposits, a key hallmark of L-ORD, and AMD (Figure 3E)." Fig 3E is not clear. Where are the apoE deposits localized? The basal side (green) seems to be localized opposite from the ApoE signal, separated by RPE nuclei. Better quality images will help to visualize the existence of sub-RPE deposits in iRPE cells from patients and define their localization. For how long was the AMPK activity inhibited to prevent the deposit formation? Is the ApoE accumulation reversible?

We have repeated the experiment and now show higher resolution images of ara-A treatment on L-ORD-iRPE (Figure 3I). In healthy-iRPE APOE is primarily apical inside the cells but in L-ORD-iRPE, we see apical signal and basal deposits trapped within the pores and on the underside of the transwell membrane. This is similar to the subretinal deposition observed in porcine and human fetal RPE cell culture models (Pilgrim et al. 2017)

This sub-RPE APOE signal (indicated by the arrow) is reduced when AMPK activity was inhibited for 1 week using Ara-A treatment. Comparing APOE accumulation with and without treatment suggests that the APOE accumulation is indeed reversible with AMPK inhibition.

- Cammisotto, P. G. & M. Bhandari (2008) Adiponectin stimulates phosphorylation of AMP-activated protein kinase alpha in renal glomeruli. *J Mol Histol*, 39, 579-84.
- Del Barco, S., A. Vazquez-Martin, S. Cufí, C. Oliveras-Ferreras, J. Bosch-Barrera, J. Joven, B. Martin-Castillo & J. A. Menendez (2011) Metformin: multi-faceted protection against cancer. *Oncotarget*, 2, 896-917.
- Dinculescu, A., S. H. Min, F. M. Dyka, W. T. Deng, R. M. Stupay, V. Chiodo, W. C. Smith & W. W. Hauswirth (2015) Pathological Effects of Mutant C1QTNF5 (S163R) Expression in Murine Retinal Pigment Epithelium. *Invest Ophthalmol Vis Sci*, 56, 6971-80.
- Fernandez-Marcos, P. J. & J. Auwerx (2011) Regulation of PGC-1 α , a nodal regulator of mitochondrial biogenesis. *Am J Clin Nutr*, 93, 884S-90.
- Ju, T. C., H. M. Chen, J. T. Lin, C. P. Chang, W. C. Chang, J. J. Kang, C. P. Sun, M. H. Tao, P. H. Tu, C. Chang, D. W. Dickson & Y. Chern (2011) Nuclear translocation of AMPK-alpha1 potentiates striatal neurodegeneration in Huntington's disease. *J Cell Biol*, 194, 209-27.
- Loubiere, C., S. Clavel, J. Gilleron, R. Harisseh, J. Fauconnier, I. Ben-Sahra, L. Kaminski, K. Laurent, S. Herkenne, S. Lacas-Gervais, D. Ambrosetti, D. Alcor, S. Rocchi, M. Cormont, J. F. Michiels, B. Mari, N. M. Mazure, L. Scorrano, A. Lacampagne, A. Gharib, J. F. Tanti & F. Bost (2017) The

- energy disruptor metformin targets mitochondrial integrity via modification of calcium flux in cancer cells. *Sci Rep*, 7, 5040.
- Milone, M. C. & U. O'Doherty (2018) Clinical use of lentiviral vectors. *Leukemia*, 32, 1529-1541.
- Monaco, A. & A. Fraldi (2020) Protein Aggregation and Dysfunction of Autophagy-Lysosomal Pathway: A Vicious Cycle in Lysosomal Storage Diseases. *Front Mol Neurosci*, 13, 37.
- Pilgrim, M. G., I. Lengyel, A. Lanzirotti, M. Newville, S. Fearn, E. Emri, J. C. Knowles, J. D. Messinger, R. W. Read, C. Guidry & C. A. Curcio (2017) Subretinal Pigment Epithelial Deposition of Drusen Components Including Hydroxyapatite in a Primary Cell Culture Model. *Invest Ophthalmol Vis Sci*, 58, 708-719.
- Qu, S., C. Zhang, D. Liu, J. Wu, H. Tian, L. Lu, G. T. Xu, F. Liu & J. Zhang (2020) Metformin Protects ARPE-19 Cells from Glyoxal-Induced Oxidative Stress. *Oxid Med Cell Longev*, 2020, 1740943.
- Rice, D. S., J. M. Calandria, W. C. Gordon, B. Jun, Y. Zhou, C. M. Gelfman, S. Li, M. Jin, E. J. Knott, B. Chang, A. Abuin, T. Issa, D. Potter, K. A. Platt & N. G. Bazan (2015) Adiponectin receptor 1 conserves docosahexaenoic acid and promotes photoreceptor cell survival. *Nat Commun*, 6, 6228.
- Sinha, D., B. Steyer, P. K. Shahi, K. P. Mueller, R. Valiauga, K. L. Edwards, C. Bacig, S. S. Steltzer, S. Srinivasan, A. Abdeen, E. Cory, V. Periyasamy, A. F. Siahpirani, E. M. Stone, B. A. Tucker, S. Roy, B. R. Pattnaik, K. Saha & D. M. Gamm (2020) Human iPSC Modeling Reveals Mutation-Specific Responses to Gene Therapy in a Genotypically Diverse Dominant Maculopathy. *Am J Hum Genet*, 107, 278-292.
- Stanton, C. M., S. Borooah, C. Drake, J. A. Marsh, S. Campbell, A. Lennon, D. C. Soares, N. A. Vallabh, J. Sahni, A. V. Cideciyan, B. Dhillon, V. Vitart, S. G. Jacobson, A. F. Wright & C. Hayward (2017) Novel pathogenic mutations in C1QTNF5 support a dominant negative disease mechanism in late-onset retinal degeneration. *Sci Rep*, 7, 12147.
- Vial, G., D. Daille & B. Guigas (2019) Role of Mitochondria in the Mechanism(s) of Action of Metformin. *Front Endocrinol (Lausanne)*, 10, 294.
- Wang, Y., G. Lu, W. P. Wong, J. F. Vliegenthart, G. J. Gerwig, K. S. Lam, G. J. Cooper & A. Xu (2004) Proteomic and functional characterization of endogenous adiponectin purified from fetal bovine serum. *Proteomics*, 4, 3933-42.
- Yavari, A., C. J. Stocker, S. Ghaffari, E. T. Wargent, V. Steeples, G. Czibik, K. Pinter, M. Bellahcene, A. Woods, P. B. Martínez de Morentin, C. Cansell, B. Y. Lam, A. Chuster, K. Petkevicius, M. S. Nguyen-Tu, A. Martinez-Sanchez, T. J. Pullen, P. L. Oliver, A. Stockenhuber, C. Nguyen, M. Lazdam, J. F. O'Dowd, P. Harikumar, M. Tóth, C. Beall, T. Kyriakou, J. Parnis, D. Sarma, G. Katritsis, D. D. Wortmann, A. R. Harper, L. A. Brown, R. Willows, S. Gandra, V. Poncio, M. J. de Oliveira Figueiredo, N. R. Qi, S. N. Peirson, R. J. McCrimmon, B. Gereben, L. Tretter, C. Fekete, C. Redwood, G. S. Yeo, L. K. Heisler, G. A. Rutter, M. A. Smith, D. J. Withers, D. Carling, E. B. Sternick, J. R. Arch, M. A. Cawthorne, H. Watkins & H. Ashrafian (2016) Chronic Activation of γ 2 AMPK Induces Obesity and Reduces β Cell Function. *Cell Metab*, 23, 821-36.
- Zhao, X., L. Liu, Y. Jiang, M. Silva, X. Zhen & W. Zheng (2020) Protective Effect of Metformin against Hydrogen Peroxide-Induced Oxidative Damage in Human Retinal Pigment Epithelial (RPE) Cells by Enhancing Autophagy through Activation of AMPK Pathway. *Oxid Med Cell Longev*, 2020, 2524174.
- Zheng, Q., Y. Yuan, W. Yi, W. B. Lau, Y. Wang, X. Wang, Y. Sun, B. L. Lopez, T. A. Christopher, J. M. Peterson, G. W. Wong, S. Yu, D. Yi & X. L. Ma (2011) C1q/TNF-related proteins, a family of novel adipokines, induce vascular relaxation through the adiponectin receptor-1/AMPK/eNOS/nitric oxide signaling pathway. *Arterioscler Thromb Vasc Biol*, 31, 2616-23.

REVIEWERS' COMMENTS:

Reviewer #1 (Remarks to the Author):

The authors have thoughtfully and adequately responded to each of my comments, as well as those of the other two reviewers. They performed well-thought-out additional experiments, including ones not suggested by reviewers, and have significantly improved the manuscript and its impact. I congratulate them on a very nice study and manuscript and am happy to accept it in its current form for publication.

Reviewer #2 (Remarks to the Author):

The authors have responded very well to all the comments in the previous review. I have no further questions.

Reviewer #3 (Remarks to the Author):

No further comments.

This manuscript is appropriate for placement in Nature Communications Biology.